# Visual barcodes for clonal-multiplexing of live microscopy-based assays

Tom Kaufman[1], Erez Nitzan [1], Nir Firestein[1], Miriam Bracha Ginzberg[2], Seshu Iyengar [3], Nish Patel[2], Rotem Ben-Hamo[1], Ziv Porat [4], Jaryd Hunter [2,5], Andreas Hilfinger [3], Varda Rotter[1], Ran Kafri[2,5✉] & Ravid Straussman [1✉]

While multiplexing samples using DNA barcoding revolutionized the pace of biomedical discovery, multiplexing of live imaging-based applications has been limited by the number of fluorescent proteins that can be deconvoluted using common microscopy equipment. To address this limitation, we develop visual barcodes that discriminate the clonal identity of single cells by different fluorescent proteins that are targeted to specific subcellular locations. We demonstrate that deconvolution of these barcodes is highly accurate and robust to many cellular perturbations. We then use visual barcodes to generate 'Signalome' cell-lines by mixing 12 clones of different live reporters into a single population, allowing simultaneous monitoring of the activity in 12 branches of signaling, at clonal resolution, over time. Using the 'Signalome' we identify two distinct clusters of signaling pathways that balance growth and proliferation, emphasizing the importance of growth homeostasis as a central organizing principle in cancer signaling. The ability to multiplex samples in live imaging applications, both in vitro and in vivo may allow better high-content characterization of complex biological systems.

[1] Department of Molecular Cell Biology, Weizmann Institute of Science, Rehovot, Israel. [2] Programme in Cell Biology, The Hospital for Sick Children, Toronto, ON, Canada. [3] Department of Chemical and Physical Sciences, University of Toronto, Toronto, ON, Canada. [4] Flow Cytometry Unit, Life Sciences Core Facilities, Weizmann Institute of Science, Rehovot, Israel. [5] Department of Molecular Genetics, University of Toronto, Toronto, ON, Canada. ✉email: ran.kafri@sickkids.ca; ravidst@weizmann.ac.il

The ability to multiplex samples has revolutionized science as well as medical practice. Genetic barcoding applications enabled unprecedented multiplexing, followed by parallel processing and analysis of dozens to hundreds of thousands of samples in applications like scRNA-Seq or functional CRISPR/shRNA/open-reading frame (ORF) screens. In contrast, in the field of image-based screens, high-order multiplexing is limited by the small number of channels that can be practically separated with common microscopy equipment. To address this, we developed visual barcodes that enable multiplexing of microscopy-based applications and used them to multiplex live-cell reporters for the study of signaling pathways dynamics in cancer cells.

To maintain growth, individual cells must balance a network that integrates information from numerous branches of signaling. In cancer, genetic and non-genetic alterations in major signaling pathways have been tightly linked to tumor initiation, progression, and response to anticancer therapies. It was also demonstrated that many of these alterations can be classified into a dozen signaling pathways, which together regulate core cellular processes such as cell fate, cell survival, and genome maintenance[1,2].

To facilitate understanding of cancer signaling, previous studies have developed genetically tagged activity reporters, i.e., fluorescent proteins that exhibit changes in abundance or localization in response to particular signaling activities[3,4]. Unlike endpoint assays, which necessitate ending the experiment in order to measure a phenotype, these reporters helped reveal the intricate dynamics of individual branches of signal transduction pathways, in live cells, at single-cell resolution. However, since it is difficult to multiplex fluorescence reporters, mutual dependencies between these separate branches of signaling remained less explored.

In this study, we describe the development and application of visual barcodes, a technology that enables multiplexing live-cell imaging applications. The visual barcodes are used as labels of the clonal identity of single cells in mixed populations. To advance the understanding of cancer signaling pathways and their crosstalk, we used visual barcodes to multiplex 12 live reporters of major signaling pathways, generating an experimental system that we term, the "Signalome".

Using the Signalome, we investigated the coordinated (multiplexed) dynamics of 12 signal transduction pathways in cancer cells that were challenged with well-characterized chemical perturbants. Our results show that multiplexing 12 reporter lines not only increases throughput but also eliminates noise associated with well-to-well variations in high-throughput drug screens. Surprisingly, our results also identify a previously undescribed binary partitioning of cancer signaling into two distinct clusters.

To maintain homeostasis, proliferating cells require mechanisms that coordinate rates of cell division with appropriate rates of biosynthesis. To sustain the proliferative demands of oncogenes, cancer cells require precisely tuned rates of biosynthesis[5]. If cells fail to double their mass between consecutive cell divisions, cell mass will progressively diminish. Conversely, rates of biosynthesis that exceed rates of cell division can result in cellular enlargement and senescence[6–8]. We show that the two clusters of signaling pathways identified by the Signalome system represent a general stress response that correlates with the cells' need to balance between growth and proliferation. Implementing Signalome in cancer cell lines thus revealed the previously underappreciated importance of growth homeostasis as a central organizing principle in cancer signaling.

Our findings thus demonstrate that the Signalome is a robust technology that can help study the dynamics of signaling pathways in a single-cell resolution and the interconnection between different signaling branches. As the system is highly modular, replacing reporters can be easily achieved to allow the study of many other questions across the different fields of biology. In addition, the visual barcodes can be used in numerous in vitro and in vivo applications in which visual deconvolution of multiplexed cell lines is of need.

## Results

**Developing visual barcodes to allow multiplexing of live imaging applications**. To construct visual barcodes, we first stably infected the A375 melanoma cell line with a lentivirus containing the nuclear marker histone 2A fused to iRFP (iRFP-H2A) to accurately demarcate nuclear and cytoplasmic compartments (Fig. 1A). Next, iRFP-H2A cancer cells were used to generate five subclones, each labeled with a cyan fluorescent protein (CFP) tagged with unique cellular localization sequences, targeting the CFP into one of five subcellular locations: nucleus, endoplasmic reticulum (ER), cytoplasm (NES), peroxisomes (Peroxi), and whole cell (WC) (Fig. 1B). CFP localization was thus used as a visual barcode that can discriminate clonal identity.

To test the accuracy of the visual barcodes, we imaged each of the five subclones separately with both phase-contrast as well as in iRFP and CFP channels. We then used CellProfiler[9] to: (1) segment the cells, (2) identify both nucleus and cytoplasmic compartments, and (3) extract texture, shape, and intensity-based features in the CFP channel for each cell (Sup code). CellProfiler Analyst[10] was used to classify single cells based on visual barcode readouts. In our implementation, we used 70% of the cells as a training set, with the remaining 30% as a validation set. The average false detection rate (false-positive barcode labeling) was 1.15%, while miss rate (false-negative rate) was 1.17%, representing very high precision and recall rates respectively (Fig. 1C and Supplementary Fig. 1A). The highest false detection rate and miss rate were both detected between nuclear and ER localizations, which were thus deprioritized from additional follow-up.

To scale up the dimensionality of visual barcodes to a 12-barcode system, we fused three of the localization signals to four fluorescent proteins (CFP, BFP, GFP, and YFP), resulting with 12 distinct visual barcodes (Fig. 1D). This increase in barcode number still maintained a very high accuracy with an average false detection rate of 1.45% and a miss rate of 1.34% (Fig. 1E and Supplementary Fig. 1B). Most detection errors were between subclones with the same fluorescence color, with the highest false detection rate detected between whole-cell and peroxisomal BFP (Fig. 1E). To explore the robustness of the system under different types of perturbations, we treated each of the subclones separately with 75 drugs (Supplementary Data 1). We found that after 48 h of drug treatment, both precision and recall were still very high in almost all drugs with only eight drugs (12%) having a miss rate of above 3%, five of which strongly affecting cell proliferation and viability. (Fig. 1F and Supplementary Fig. 1B, C).

Note that one option to further increase the number of barcodes is to combine two barcode proteins into a single cell. By using four fluorescent proteins and five localizations, up to 160 different visual barcodes can be generated. Indeed, we were able to demonstrate a very high precision and recall rate for barcode calling even when barcodes with the same localization but different colors were combined (Supplementary Fig. 1D, E).

Lastly, we showed that the clones can also be separated when cells are in suspension, using the Imagestream high-resolution microscopy and flow cytometry system (Fig. 1G–J). As our system lacked a laser to detect YFP, we only multiplexed nine of the subclones. To demonstrate that the visual barcodes can also be used in vivo, we mixed the nine clones and implanted them subcutaneously in a nude mouse. When the tumor reached a

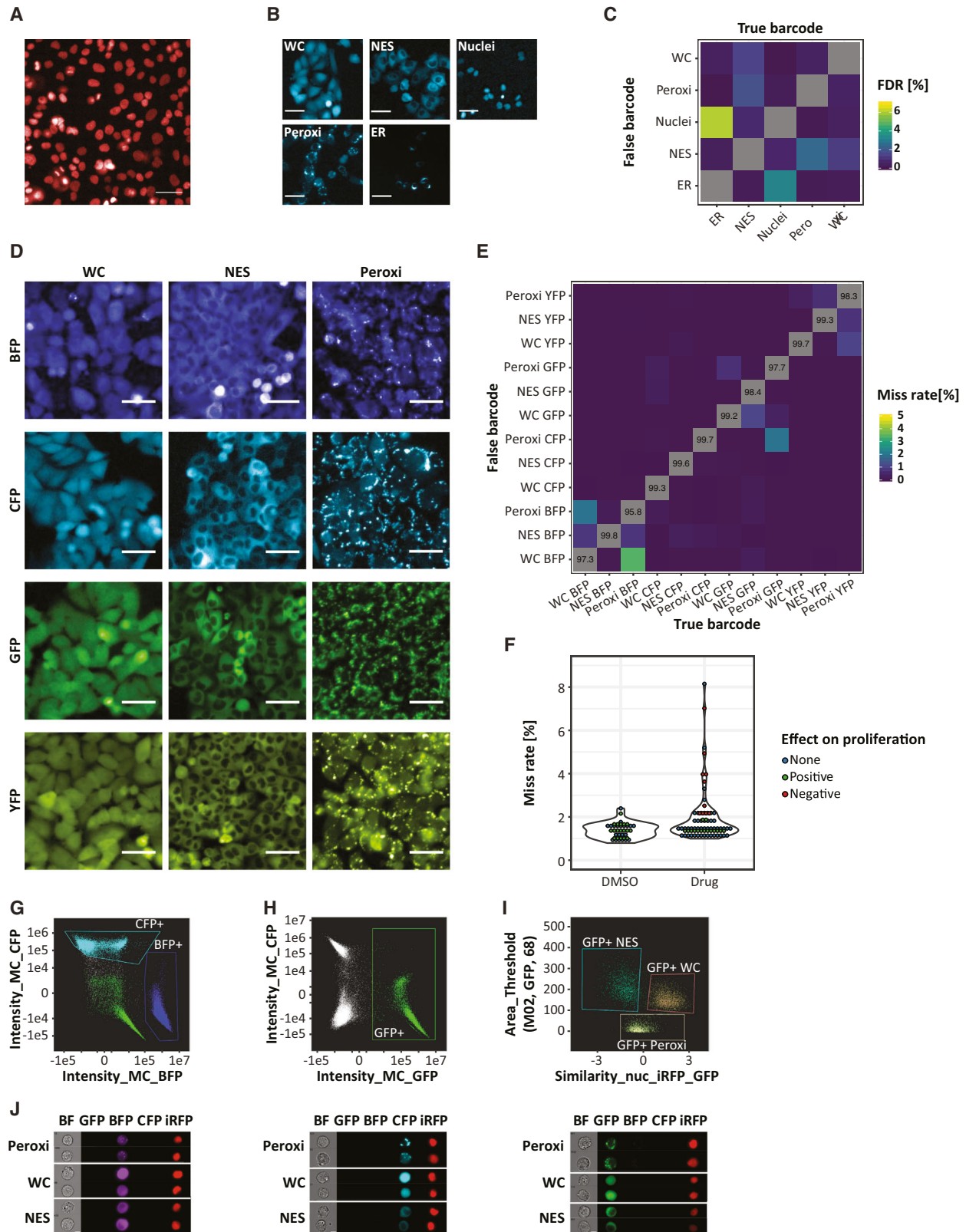

diameter of 8 mm it was excised, dissociated into single cells, and analyzed by the Imagestream system, demonstrating that all nine clones could be detected (Supplementary Fig. 1F–J).

**Generating the "Signalome" reporting cell lines**. To generate the Signalome, we assembled 12 previously published and well-characterized reporter constructs (Supplementary Fig. 2A), each

associated with the activity of a different cancer-related signaling pathway. To enable their multiplexing, we replaced the original tagged fluorescent protein in each of the 12 reporter vectors with the mStrawberry fluorescent protein. We used two different types of reporters: reporters that drive the expression of the fluorescent protein by a specific transcription response element (TRE) or kinase translocation reporters (KTRs) that translocate the

**Fig. 1 Developing visual barcodes for multiplexing live imaging applications. A** Representative image of the A375 cell line with an iRFP-H2A nuclear marker. **B** Representative images of five CFP localizations in the A375 cell line: Whole Cell (WC), Nuclear Export Signal (NES), Nuclei, Peroxisome (Peroxi), and Endoplasmic Reticulum (ER). **C** Heatmap of false detection rate (FDR) for barcode calling for the five CFP localizations in the A375 cell line. **D** Representative images of all 12 visual barcodes used in A375 cell line. **E** Average miss rate of barcode calling for all 12 visual barcodes of the A375 cell line, treated with DMSO controls ($n = 39$). Numbers in the diagonal represent the sensitivity for each barcode. **F** Violin plots showing miss rate for all 12 visual barcodes of the A375 cell line, treated with DMSO controls ($n = 39$) or with drugs ($n = 75$). **G–I** Scatter plot showing the separation of nine A375 clones with visual barcodes by the ImageStream system according to their fluorescent color and localization. The separation by localization is only demonstrated for GFP-positive clones (**I**). **J** Representative images from ImageStream of all nine clones. Two cells from each clone are presented. Scale bar in (**A** and **B**) is 50 μm, in (**D**) 100 μm and in (**J**) 7 μm. Source data are provided with this paper.

fluorescent protein from the nucleus to the cytoplasm upon activation of upstream signaling (Supplementary Fig. 2A).

We then infected each of the 12 A375 subclones that have visual barcodes with one of the 12 reporters (Fig. 2A). Next, we validated that the proliferation rate of the single-cell derived reporter subclones is not different from that of the parental cell line (Supplementary Fig. 2B). We also validated that all 12 subclones are as sensitive to the BRAF inhibitor, vemurafenib, as the parental A375 cell line (Supplementary Fig. 2C). Lastly, we pooled together all 12 subclones generating the A375 signalome cell line, and demonstrated that the proportions of the 12 different clones remained constant over 48 h of culture (Fig. 2B).

An advantage of single-cell measurements is that perturbations can be characterized not only for their influence on population average but also on the full distribution (i.e., the frequency of cells with low, medium or high signaling activity). To quantify condition-dependent differences in the distribution of reporter activity, we used the Kolmogorov–Smirnov (KS) test as it is a nonparametric test that can also detect changes in the distribution that are not reflected by the mean of the distribution (Supplementary Fig. 2D, E)[11,12]. To add to the KS score, activity scores were assigned a positive or negative sign based on the change in direction of the mean of these distributions. We used a threshold of $\pm 0.2$ to mark a significant change in activity as it represents the 0.5 percentile of reporter activity across our DMSO control wells (Supplementary Fig. 2H). To validate the reporters' activity, we first used known positive or negative regulators for each of the reporters demonstrating a perfect correlation between the reporter's score and the expected drug effect (Supplementary Fig. 2F). We also compared the A375 signalome cell line activity scores to the activity of all 12 reporters when tested separately and found a very strong correlation between the two ($r = 0.75$, $P < 10^{-16}$) (Fig. 2C). As expected, we also found that Vemurafenib and the MEK inhibitor Trametinib, both inhibitors of the MAPK pathway, exerted highly similar effects on the A375-Signalome cell line (Fig. 2D–F). Finally, to validate our reporters' activity using different methods, we used western blot analysis demonstrating full agreement between both techniques (Supplementary Fig. 2G).

An advantage of multiplexed, time-dependent measurements on signaling is the ability to differentiate direct and indirect drug influences. For example, while MAPK inhibitors (vemurafenib and trametinib) promoted detectable changes in measured ERK signaling that were observed 1.5 h into drug treatment, an influence of these same drugs on other pathways was also observed, albeit at much later times (Fig. 2D–F). This is in agreement with the direct effect of these drugs on the MAPK pathway and the subsequent adaptive response of the other pathways to the inhibition of the MAPK pathway. Indeed, previous reports already described activation of PKA[13], NFkB[14], HIF[15], and YAP/TAZ[16] in response to vemurafenib and suggested that these adaptations can contribute to resistance to MAPK inhibition. In addition, we observed a significant upregulation of retinoic acid receptor activity 48 h after BRAF

or MEK inhibition ($P$ value $< 6 \times 10^{-5}$). Interestingly, examination of three independent cohorts of melanoma patients demonstrated that patients with high activity of the RAR/RXR pathway, as calculated from expression data by PathOlogist[17,18], had an overall better survival (Supplementary Fig. 2I–K). Therefore, it may be of interest to continue and better explore the role of RAR in melanoma and its response to therapy. These measurements also demonstrate the efficiency and throughput of the Signalome system: with one 384-well plate, we screened the influence of 75 drugs (in triplicates) and 39 DMSO controls on 12 branches of signaling, in multiple time points, and at single-cell resolution. A single signalome plate thus provided measurements on half a million cells from each of the different time points.

To demonstrate that the visual barcodes and a signalome system can be readily applied to other cell lines we generated two more signalome cell lines using the PC9 EGFR-mutated non-small-cell lung cancer cell line and the SK-MEL-5 BRAF-mutated melanoma cell line. We were able to demonstrate that our barcode precision and recall rates are also very high in these cell lines (Supplementary Fig. 3A–D) and that an early inhibition of ERK by both inhibitors can be detected followed by cell-line-specific adaptive mechanisms. For example, while inhibition of BRAF or MEK in A375 melanoma cell line resulted in activation of the YAP/TAZ pathway, the effect of BRAF/MEK inhibition in the SK-MEL-5 cell line resulted in inhibition of the YAP/TAZ pathway (Supplementary Fig. 3E–H). Here, again we found that EGFR inhibition in the PC9 cells also drives upregulation of RAR activity.

**Large-scale correlations in signaling suggest a generalized response that is compound independent.** To investigate interdependencies in the cancer signaling, we treated the pooled A375 signalome cell line with a library of 422 well-characterized chemical perturbants (Supplementary Data 2, Supplementary Data 3, and Supplementary Fig. 4A). Of all tested compounds, 122 (28.9%) promoted significant changes in at least one of the reported pathways (KS absolute score $> 0.2$) (Fig. 3A, Supplementary Fig. 4I, Supplementary Data 4, and Supplementary Data 5). As expected, different drugs with similar targets displayed similar patterns of reporters' changes in response to these drugs (Supplementary Fig. 4B–G).

Surprisingly, in addition to target-specific signatures, unsupervised clustering of the reporters' activity scores also suggested a higher structure that partitions the signal transduction signatures into three clusters, two of which seem anticorrelated (Fig. 3A and Supplementary Fig. 4I). Cluster A groups compounds that seem to all activate the pathways PKA, AKT, ERK, p38, and JNK while inhibiting WNT, p53, NFkB, RAR, HIF, and YAP\TAZ; while cluster B contains drugs that orchestrate the opposite response. Cluster C contained drugs that did not follow this dichotomy. The large number of drugs with varying mechanisms of action that result in signaling clusters A and B, suggests a coordinated response to drug treatment that involves all of our measured branches of signaling and is surprisingly

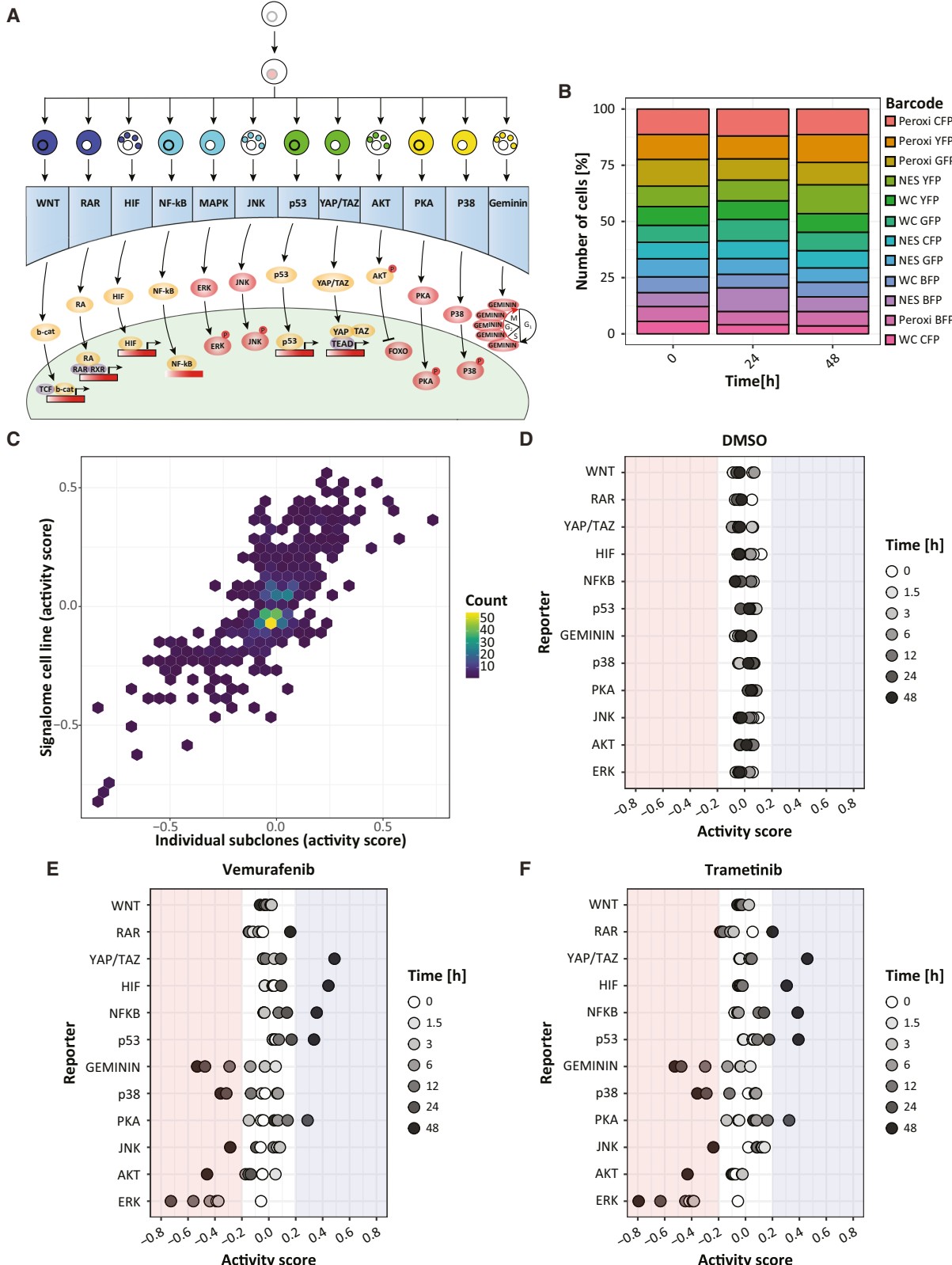

independent of drug target. As a case in point, Fig. 3B shows a negative correlation between the activities of p53 and p38 which is persistent across a wide diversity of chemical perturbations. Drugs that diminished p38 activity correlate with equivalent increase in the activity of p53 and vice versa (Pearson's $r = -0.517$, $P < 2.2 \times 10^{-16}$) (Fig. 3B). More generally, drugs promoted positive correlations among pathways within cluster A

or B (intracluster correlations) and negative correlations when comparing pathways from cluster A to pathways from cluster B (intercluster correlations). For simplicity, we will refer to these two clusters as the *p38 signaling state (cluster A)* and the *p53-signaling state (Cluster B)* (Fig. 3A).

Chemical perturbations can generate correlated influences by simultaneously affecting more than one target. Such correlated

**Fig. 2 Generating the A375 "Signalome" reporter cell line. A** Illustration of the 12 clones that were used to generate the A375 Signalome cell line. A375 cells were first infected with iRFP-H2A to mark the cell nucleus. Then, 12 clones were generated with 12 visual barcodes. Lastly, a different live reporter was added to each of the clones. Transcription activating reporters are represented by gold while translocation reporters are represented in red. Binding partners in the nucleus are represented in purple. **B** Relative number of cells from each clone in a DMSO control wells over time. **C** Scatter plot showing the correlation between the reporter activity scores, for all 12 clones when grown separately or as part of the Signalome cell line, in response to 75 drugs. **D–F** Reporter activity plots of the A375 signalome cell line in response to DMSO, vemurafenib (1 μM) or trametinib (0.125 μM) over time. Blue and red backgrounds represent activation or inhibition scores above 0.2 or below −0.2, respectively. The average cell count per reporter in (**D–F**) is: 656, 545, and 530 cells, respectively. Source data are provided with this paper.

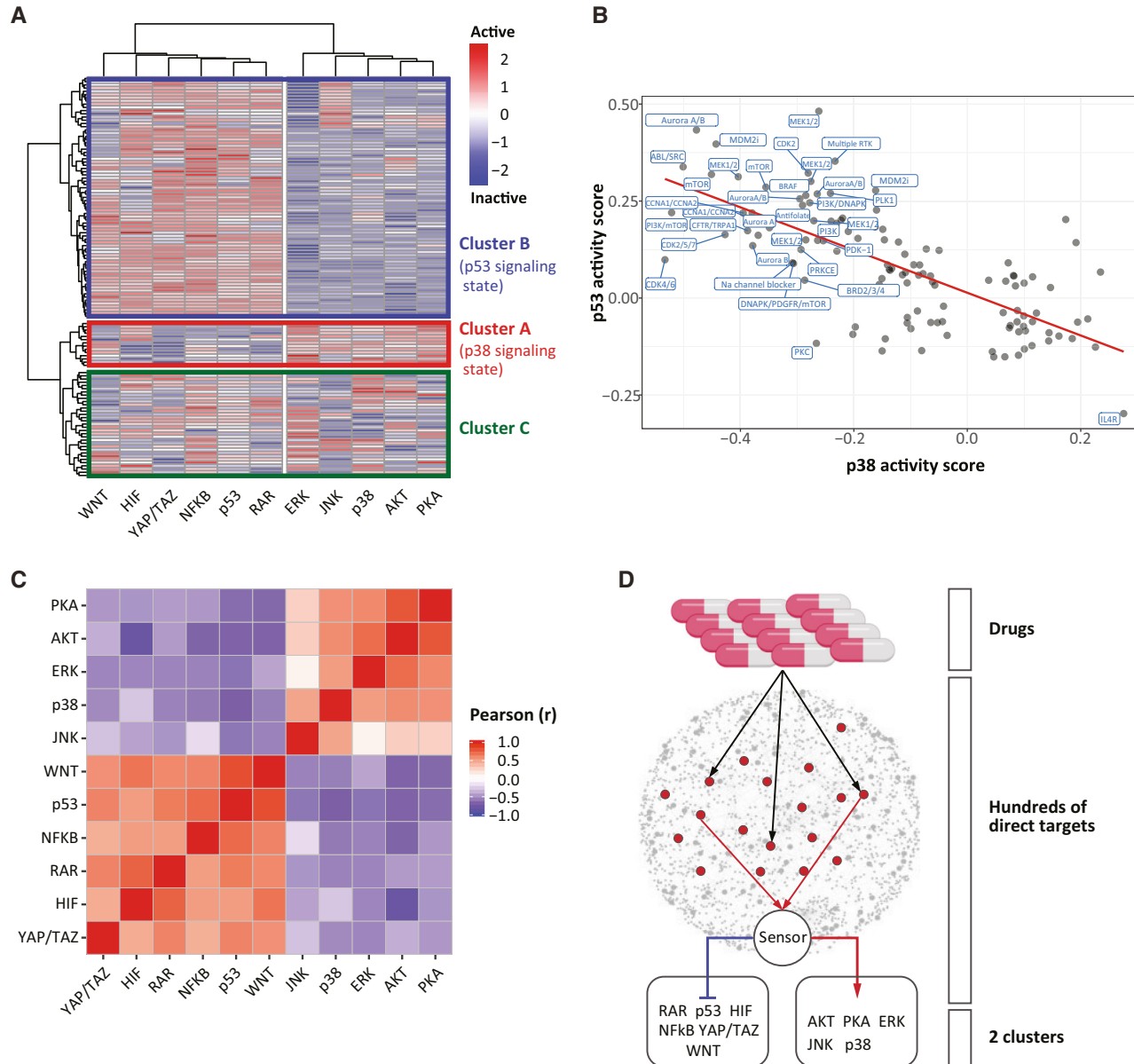

**Fig. 3 Large-scale correlations in signaling suggest a generalized response that is compound independent. A** Unsupervised hierarchical clustering of A375 signalome cells treated by 122 active drugs according to their activity scores. Due to technical error, Geminin reporter was not measured for all drugs and was thus discarded. Results from all 12 reporters can be seen in Supplementary Fig. 4I for the subset of drugs that for which Geminin data are available. **B** Scatter plot showing the correlation between the activities of the p38 and p53 reporters after 48 h of treatment with 122 active drugs. **C** Heatmap showing the pairwise Pearson correlations between the different A375 Signalome clones, after 48 h of treatment with 122 active drugs. **D** A model proposing how drugs with different mechanisms may converge into two major signaling states. While each of the drugs has different targets, many of the targets affect the same sensing mechanism that later governs the p53- vs p38-signaling states. Source data are provided with this paper.

influences, however, should be compound-specific, relating to target affinities that differ from one compound to another. By contrast, we found that the same pairs of pathways are positively, or negatively correlated, across a large number of drugs ($n = 122$) that have multiple and highly different targets. (Fig. 3B, C). To demonstrate that this partition is not cell line-specific, we treated the PC9 signalome cell line with 247 drugs and found very similar bifurcation into two anticorrelated clusters of pathways (Supplementary Fig. 4H and Supplementary Data 6). The question, therefore, is as follows: how can such a wide variety of perturbations, each associated with different targets, converge onto only two main outcomes? We reasoned that the partitioning of the signaling pathways into two clusters suggests a regulatory process that is common and upstream of all of our measured branches of signaling (Fig. 3D).

**Large-scale correlations in signaling are present pre-treatment and increase over time by multiple drugs**. To gain insight into the nature of this upstream regulatory process, we first performed time-course measurements to ask, how soon after drug treatments do the pairwise correlations become apparent? We found that pairwise correlations in pathway activity became more and more prominent throughout the 48 h following drug treatments (Fig. 4A, B). The segregation of the two signaling states post drug treatment is also apparent from visual inspection of time-course measurements (Fig. 4C–H). Trajectories of activity of the reporters in response to six representative drugs that are chemically distinct and associated with different targets, demonstrated that while three drugs promoted the p38 signaling state (Fig. 4C–E), the other three activated the p53-signaling state (Fig. 4F–H). Altogether, these results demonstrate the binary partitioning of the signaling pathways by showing that a variety of different drugs, each associated with different targets, converge to promote two main signaling states outcomes.

These results suggest that the p38- and p53-signaling states are mediated by a process that, in response to drug treatments, gradually increases its influence, or activity, over time. Since it is likely that the drug treatments only promoted the activity of an already existing process, we were curious as to why the correlations shown in Fig. 3C seem absent in unperturbed cells (Fig. 4B)? One possibility is that, prior to drug treatments, signaling pathways are subjected to the simultaneous influence of several competing regulatory demands, each pulling in a different direction. According to this interpretation, drug treatments promote correlated signaling by increasing the relative weight of one particular regulatory process—most likely a stress response —such that its effect is no longer averaged by competing influences. This model suggests a testable hypothesis: the binary partitioning of the 12 pathways should become apparent in untreated cells if the measurements are normalized for independently existing correlations.

To this end, we used Principal component analysis (PCA), a technique that transforms a dataset into a linear combination of independently existing multivariate correlations. As expected, PCA confirmed the high degree of correlations by identifying a single principal component (PC1) that explains almost 50% of the variance after 48 h of treatment (Fig. 5A). Further, the first principal component clearly identified the two signaling states; drugs that promote the *p38-signaling* or *p53-signaling* states are characterized by negative or positive values of PC1 respectively (Fig. 5B). Next, we repeated the PCA, but this time on measurements collected prior to drug treatments (time zero). Note that in this latter implementation of PCA, variation in measured activities did not reflect differences in drug response, as no drugs were yet applied, but rather, small well-to-well

variations like differences in evaporation rate or oxygen concentrations. In early landmark studies[19,20], such well-to-well (or sample-to-sample) variation in cell division rates and in cell size served as pivotal evidence for the functioning of cell size checkpoints in animal cells[21]. Since the signalome provides measurements on 12 pathways in each well, it can test whether these small variations will lead to a well-specific shift in the signaling states that can be detected by PCA. Indeed, PCA significantly identified both p38- and p53-signaling states also in the unperturbed cells (Fig. 5C).

To further explore whether the p38- and p53-signaling states precede drug treatments, we tested whether measurements on cells that were not exposed to drug treatments can predict the specific correlations observed post drug treatments (Supplementary Fig. 5). This analysis identified dynamics that are classically characteristic of homeostasis (Fig. 5D). In the first hour post drug treatment, chemical perturbations effectively eliminated correlated activities that linked the different branches of signaling in the unperturbed cells. Several hours into drug treatment, however, correlated activities resumed and, in fact, gained more prominence. These results suggest a general stress program that: (A) had functioned in cells that were not subject to drug treatment (B) was effectively eliminated in the first hour of drug treatment and (C) had resumed activity in the hours following drug treatment. This observation can also be visualized in Fig. 4C–H, in which the two signaling states are apparent at time zero (pre-drug treatment), lost at one hour, and gains significance at the latter time points.

**The p38- and p53-signaling states are linked to perturbations in cell size**. The constancy of the two signaling states, in the face of diverse chemical perturbations, suggests that the large-scale correlations described by these signaling states function to support some process that is critical in our cell lines. Since our measurements were performed on cancer cells, we further reasoned that the process in question may relate to demands imposed by continuous cell divisions. To maintain homeostasis, proliferating cells must double their mass between consecutive cell divisions. In cancer, this requirement may be more critical[22–24]. If cancer cells fail to match the proliferative demands of an oncogene with equivalent increases in biosynthesis, cell size will decrease over time[21]. We therefore wondered whether the observed pattern of coordinated signaling results from stress-sensing systems that respond to changes in cell size resulting from imbalances in cell growth and cell division (Fig. 6A).

As a first step, we asked whether drugs that selectively interfere with rates of biosynthesis trigger compensatory mechanisms that will help the cell to reach a new steady-state between growth and proliferation rates. We first used the Retinal epithelial line RPE1, an hTERT immortalized non-cancerous cell line, as it is highly suitable for studies of cell size[21]. To inhibit cell division rate, we used various chemical inhibitors of Cyclin-dependent kinases (CDK) like SNS-032[21] while to lower biosynthetic activity (cell growth), we either inhibited protein synthesis by Cycloheximide or mTOR activity by Rapamycin and Torin. In all cases, drug doses were carefully optimized to ensure that cells are still proliferating and are not undergoing complete cell cycle arrest. For quantitative measurements of cell growth (protein synthesis per unit time), we followed a previously described protocol for single-cell measurements of total macromolecular protein mass using fluorescently labeled succinimidyl ester (SE) that label all proteins (Supplementary Methods)[21,25,26].

Our results demonstrate that during the initial hours of Rapamycin treatment, while cell growth was rapidly inhibited, the rates of cell division were relatively unaffected (Fig. 6B).

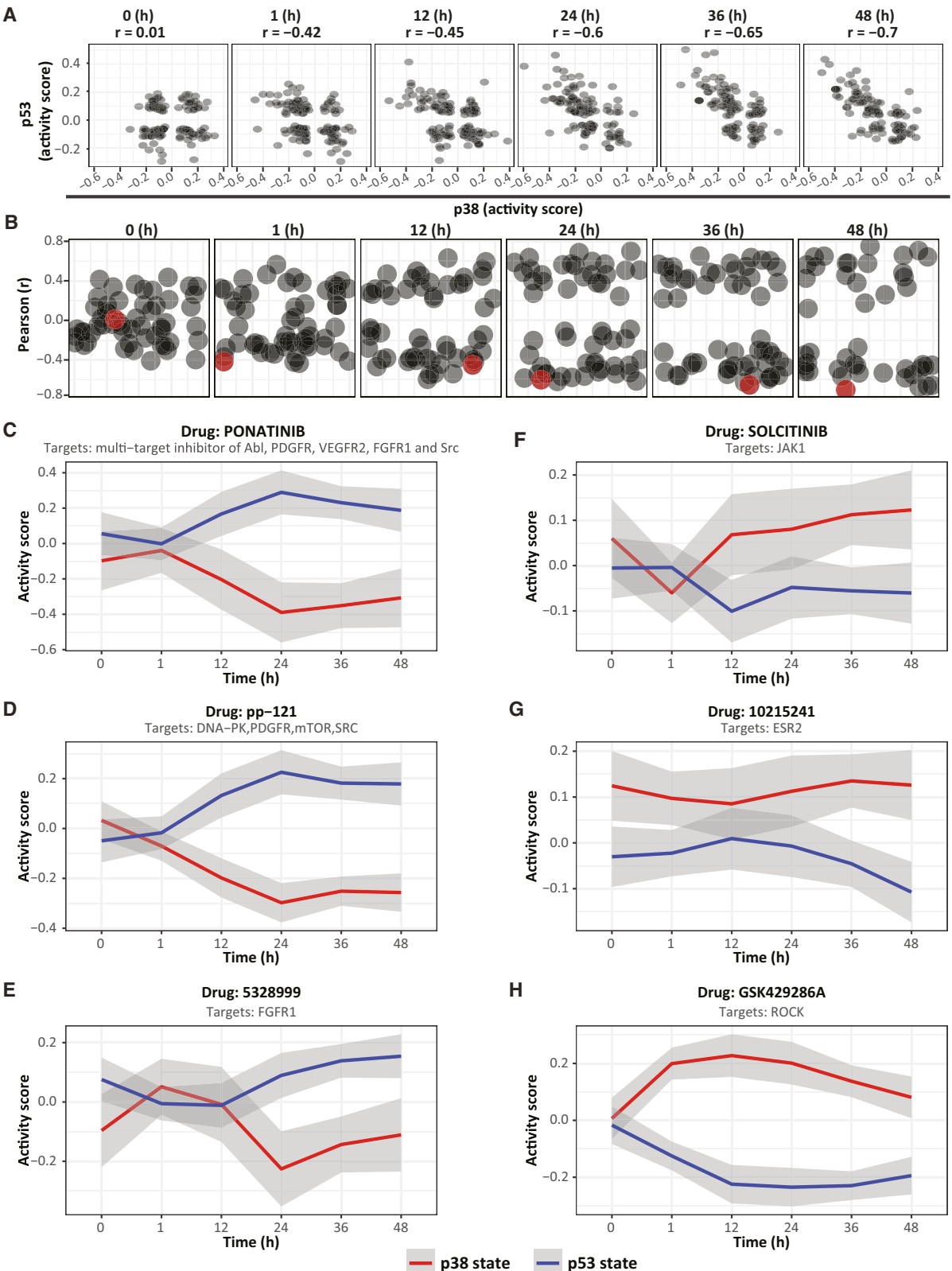

Conversely, the CDK2 inhibitor SNS-032 lowered rates of cell division but did not affect cell growth (Fig. 6B). At the later time points, however, a coordination of growth and proliferation was re-established, but at a slightly different setpoint. Cells with inhibited rates of biosynthesis adapted by promoting longer periods of biosynthetic activity (longer cell cycles). Yet, this lengthening of the cell cycle fell short of a perfect adaptation, resulting in paired values of growth and division that fell slightly below the proportionality line. Similarly, to adapt to the longer growth periods imposed by CDK2 inhibitors, cells lowered the amount of protein synthesized per unit time. Yet, here too, the compensation was incomplete, resulting in paired values that lay above the line. Extending these results to multiple cell cycle or cell growth inhibitors across five cell lines demonstrated that

**Fig. 4 Drug treatments increase the correlations between the activity of pathways. A** Scatter plots of the activity scores of p38 and p53 reporters in the A375 signalome cell line before and at multiple time points after treatment with 122 active drugs. Each dot represents a different drug. Pearson's *r* is depicted for each of the time points. **B** Pearson correlation coefficients were calculated for each pair of pathways in the A375 signalome cell line before and at different time points after treatment with 122 active drugs. Each datapoint represents a correlation value for one given pair of pathways over all 122 active drugs. The correlation between the activity score of p38 and p53 pathways is marked by a red circle. **C–H** Locally weighted smoothing (Lowess) regression of all reporters in each of the two signaling states is shown for six representative drug treatments, each associated with different drug targets. While three of the drugs drive the p53-signaling state (**C–E**), the other three drugs drive the p38 signaling state (**F–H**). The bold (red and blue) lines and the gray sleeves represent mean values and +/− SEM respectively. Source data are provided with this paper.

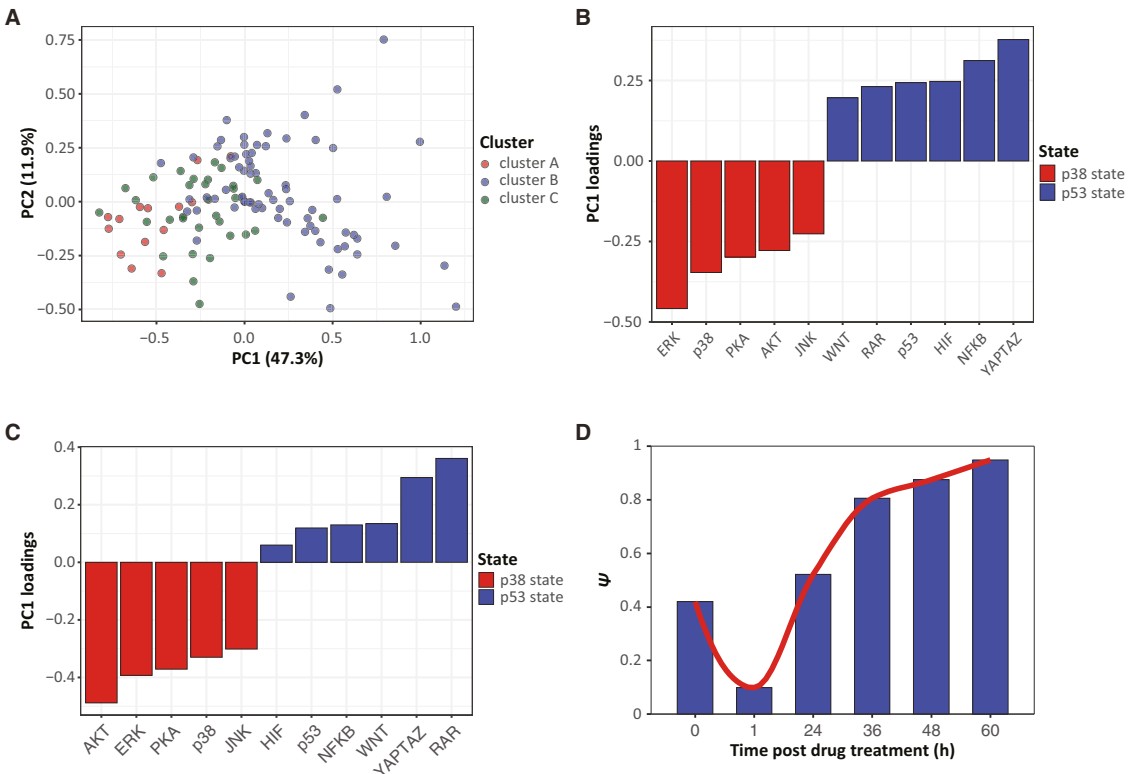

**Fig. 5 PCA suggests that the p38 and p53-signaling states exist pretreatment and increase in weight over time. A** PCA of the activity scores of 11 signaling pathways after 48 h of treatment with 122 drugs. The color of each drug is indicating its cluster in Fig. 3A. **B, C** Bar plots representing the PC1 loading of each pathway after 48 h of drug treatment (**B**) or pretreatment (**C**). **D** Variance of pathway activity, when projected on the principal components calculated from measurements on cells that were not exposed to drug treatment. Source data are provided with this paper.

incomplete compensation of the growth and proliferation rate is a general phenomenon (Fig. 6C). In conclusion, our results suggest that: (A) Drugs that perturb rates of biosynthesis trigger compensatory changes in division rates, and vice versa; (B) The adaptation of growth rates and division rates to drug treatments is typically incomplete, resulting in paired values of growth and division rates that lie above and below the proportionality line.

To investigate the possibility that the p38- and p53-signaling states that we observed are related to the homeostasis of cell size, we asked whether drugs that induce these two states differ in their influence on cell size. To that end, we scored each compound for the extent that it promoted the p38-signaling vs p53-signaling states (Supplementary Methods). We then used the signalome single-cell resolution measurements to calculate the influence of each drug on cell division rate and cell size. Consistent with our hypothesis, we found that drug treatments that promote the p38 state correlated with a smaller cell size while drugs that promote the p53 state correlated with increased cell size (Pearson's $r = −0.625$, $P = 1.68 \times 10^{−23}$) (Fig. 6D).

To further strengthen our model, we asked whether drug targets that promote the p53- and p38-signaling states were

previously characterized as regulators of cell-size homeostasis. In mammalian cells, a comprehensive screen for mechanisms of cell-size homeostasis was described in Liu et al.[25]. In that study, highly quantitative cell-size measurements were performed on cells that were interrogated with the NIBR MoA drug library[27]—a highly characterized chemical library designed to interrogate mechanisms of action. To extract target-specific cell-size influences, we implemented a linear regression model (Supplementary Methods). This resulted with a list of genes, each with a predicted influence on rates of cell division and on cell size. Figure 6E reveals a significant correlation between the influence of drug targets on cell size (as quantified in Liu et al., Supplementary Data 7). Specifically, targets that were characterized in Liu et al. as inhibitors of cellular growth were independently identified in our study as targets that promote the activation of the p38-signaling state. Conversely, targets characterized by Liu et al.[25] to slow the rates of cell division were independently identified in our study as targets that promote the activation of the signalome reporters associated with the p53-signaling state. Notice, that while the signalome data were produced on A375 melanoma cells, the data of Liu et al.[25] was generated using HeLa cervical cancer cells.

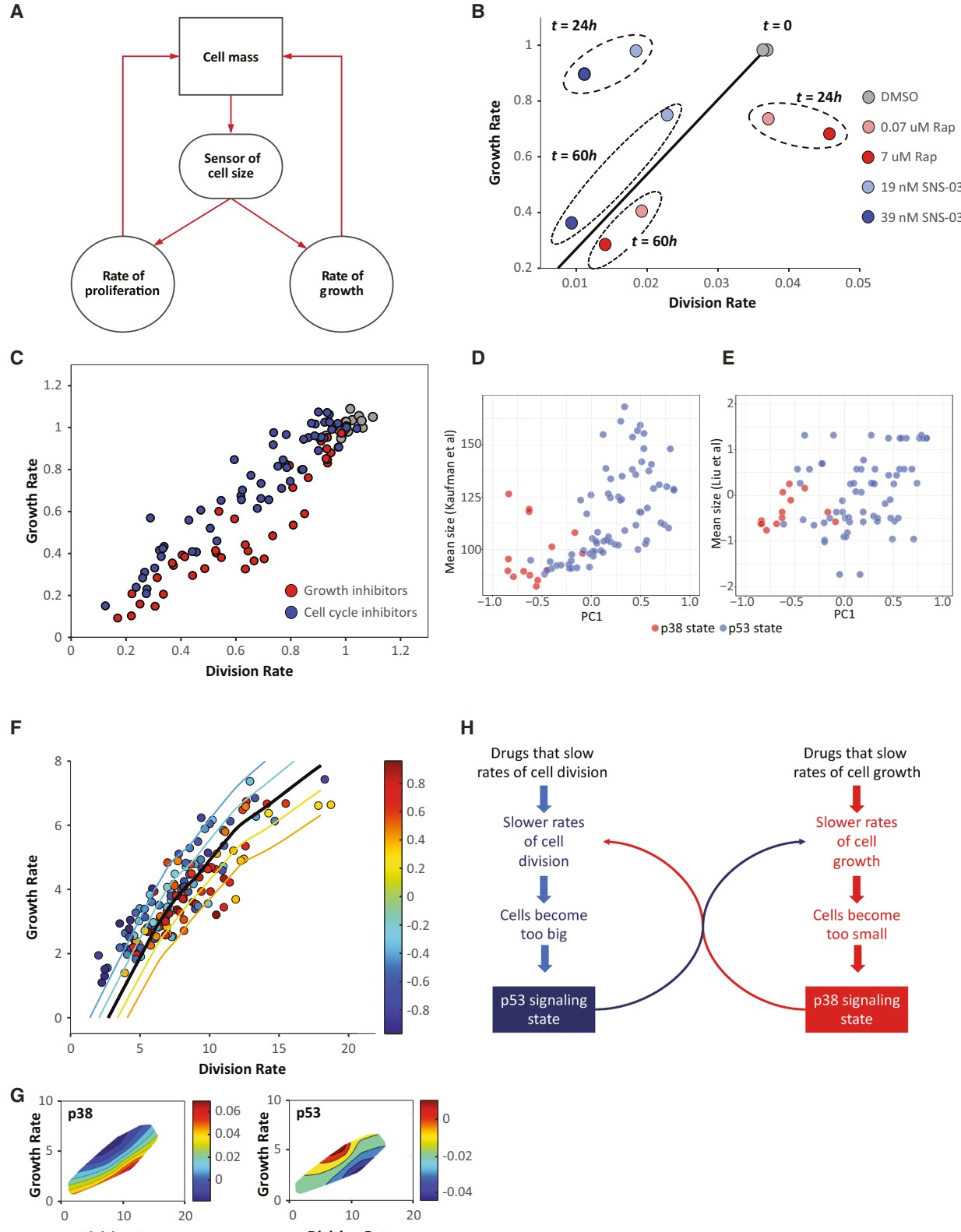

These correlations between signal transduction measurements on A375 cells and cell-size measurements on HeLa cells indicate that the link of the two states with cell size is common across different cell lines.

Next, we used the signalome to test whether the two signaling states correlate with imbalances in cell growth and division rates. We found that drugs that disproportionately decrease growth rates (i.e., data points below the diagonal) were associated with the p38-signaling state, while drugs that disproportionately decrease proliferation (i.e., points above the diagonal) induced the p53-signaling state (Fig. 6F, G).

Additional support for the distinct association of the p38- and p53-signaling states with cell growth and cell division came from functional annotation enrichment analysis using the STRING

**Fig. 6 Cell growth and proliferation are tightly regulated and correlate with p38- and p53-signaling states. A** A model demonstrating how sensing of cell size can affect both cell growth and proliferation to keep homeostasis of cell size. **B** Scatter plot showing the initial and long-term effects of rapamycin or SNS-032 on the average cellular growth rate and division rate of Rpe1 cells. Data points indicate the average growth and division rates measured: (1) during the first 24 h of drug treatment and (2) during 24–60 h of drug treatment. **C** Average growth rate vs. division rate in five cell lines (Rpe1, HeLa, U2OS, SAOS2, 16HBE) treated with either growth inhibitors (red) or cdk1/2 inhibitors (blue). Measurements in each cell line were normalized by the values measured for untreated control samples (gray) of the same cell line. The growth inhibitors used were: Cycloheximide, Torin-2, and Rapamycin, at varying doses (de- tailed in "Methods"). The cdk1/2 inhibitors used were: SNS-032, PHA848125, Cdk2 Inhibitor III, and Dinaciclib, at varying doses (detailed in "Methods"). **D, E** The average cell size for a given drug correlated with its PC1 value calculated on the reporters' activity for data from Kaufman et al. (**D**) and Liu at al. (**E**). **F** Average growth rate vs. division rate for A375 cells in Signalome screen. Each circle represents one screened condition (drug treatment). The circle's color indicates the value of PC1 in that condition. Contour lines show the average value of PC1 as a function of growth rate and division rate. **G** The average level of p38 (top) and p53 (bottom) activity as a function of growth rate and cell cycle length. **H** A model proposing how drugs which affect cell division activate the p53-signaling state while drugs that affect cell growth activate the p38-signaling state. Each state in return actives a compensation mechanism resulting in a new equilibrium. Source data are provided with this paper.

database (http://string-db.org/)[28–31]. Relying on the list of the drugs that selectively promote the p38- or p53-signaling states, we asked whether drug targets that promote the p38- or p53-signaling states are functionally distinct. Indeed, we found that while drugs that promote the activation of the p53-signaling state were enriched for inhibitors of the cell division cycle, drugs that promoted the activation of the p38 state were significantly enriched for regulators of anabolic activity, including the Insulin/mTORC1 pathway, as well as regulators of glucose, glycogen and carbohydrate metabolism (Supplementary Fig. 6A, B).

In cancer, disproportional changes in growth and division can spontaneously result from intracellular genetic changes or from external stresses, including nutrient or growth factor deprivation. To test if the p38- and p53-signaling states are represented in human cancers, we mined the TCGA (https://www.cancer.gov/tcga) to retrieve proteomic measurements from 8167 human tumors that span 32 different types of cancer[32,33]. To compare signalome reporters with TCGA, we assembled a list of eight proteins or phosphoproteins that are known to correlate with the activity of pathways included in the signalome. Using these proteins as a surrogate for pathway activity we separately analyzed each of the 32 cancers for correlated signaling. Specifically, for each cancer, we calculated all pairwise-correlation coefficients related to the signalome pathways. Our results demonstrated that, with the exception of cholangiocarcinoma, similar signaling bifurcation was present in all of the other cancer types as demonstrated in Supplementary Fig. 6F. This further indicates a common regulatory process upstream of all of our measured branches of signaling (Fig. 3D).

As a final question, we asked what is different about the drugs in cluster C (Fig. 3A) that allows them to escape the mutually exclusive association with either the p53- or the p38-signaling states? One possible hypothesis is that drugs in this cluster target mechanisms that are essential for the maintenance of cell-size homeostasis by uncoupling the cellular feedback mechanisms that balance cell proliferation and cell growth. In this context, it is intriguing to point out that CDK4 inhibitors are among the highest-scoring drugs in this group. Indeed, recently, we and others have shown that cell-size homeostasis critically depends on the CDK4[34] (but not CDK2) which may thus explain why CDK4 inhibitors do not adhere to the p38- pr p53-signaling states (Supplementary Figs. 6C–E and 7). Further studies are needed to better understand the mechanisms that drive other drugs to cluster C.

## Discussion

Live reporters are widely used to study signaling dynamics in cells. However, currently, the ability to multiplex live reporters together is limited by the number of reporters that can be multiplexed and the ability to follow multiplexed reporters for

multiple days rather than hours[35]. The integration of information from multiple signaling branches is critical for the understanding of complex biological processes. In this study we introduced visual barcodes, a fluorescent protein coupled to a specific subcellular localization peptide, which allows multiplexing cells in live imaging applications. We demonstrate that visual barcodes are robust to perturbations, have a high precision and recall rates and are applicable for multiplexing both in vitro and in vivo. Multiplexing of different subclones not only increases the throughput of experiments but also reduces cost and well-to-well or animal-to-animal variation[36]. Adding more fluorescent proteins or cellular localizations to the system can augment its multiplexing potential, and we predict that the system can be easily expanded to 20-plex combinations if only one barcode is used per cell and more than a 200-plex if two visual barcodes are used per cell. Deconvolution of the visual barcodes was done using freeware, thus enabling the use of the system without licensing limitations. The visual barcodes system can be used for a very wide variety of applications such as competition assays between clones with different perturbations in vitro or in vivo, live tracking of cells with reduced risk of switching between subclones, as well as multiplexing of live reporters, as we demonstrated by the Signalome cell lines.

For generating the Signalome cell lines, we added a different fluorescent reporter for each of our 12 validated visual barcodes subclones, reporting for major signaling pathways in cancer cells. While we generated each of the Signalome subclones using three consecutive rounds of infections (nuclear marker, visual barcode, fluorescent reporter), we envision that a visual barcode and a reporter could be integrated into a single plasmid thus allowing one round of infection with a mix of plasmids on a cell line with a nuclear marker, allowing the generation of additional Signalome cell lines in days rather than weeks.

While the signalome technology allows multiplexing live reporters in the same cell (Supplementary Fig. 1D, E), adding more than a very small number of such reporters per cell may have a substantial effect on the cells. Indeed, to the best of our knowledge, previous works that combined multiple live reporters in a single cell, have not exceeded a maximum of five reporters per cell[35,37–40]. To achieve a much greater multiplexing, we mixed clones of cells, each containing a single reporter. While we demonstrate the use of 12-plex signalome cell lines we believe that much higher multiplexing can be easily achieved by this technology. In addition to the advantage in throughput by hi-plex reporter cell lines, as the readout of all reporters comes from the same wells, a substantial reduction in well-to-well variability is expected when comparing between the effect of perturbations on the different reporters. On the other hand, there are clear limitations for not comparing between reporters that are present in the same cells. The most obvious limitation is the difficulty to

differentiate between events that occur in the same cell or in different cells. For example, if following a perturbation a subset of the cells of two clones that report on two signaling pathways show inhibition of the activity in these pathways, follow-up experiments will need to be carried out to find out if inhibition of both pathways always accrue together in the same cells or maybe in different cells. In addition, as our signalome clones were frequently expanded from single cells, it is important to validate that the main findings also apply to the whole population and do not reflect a cell-specific phenotype. One way to overcome this limitation is that each signalome reporter cell line will originate from multiple cells rather than from a single one. It remains to be seen what effect of such an approach will have on the recall and precision rates of the visual barcodes.

To better understand the interdependencies of signaling pathways, we treated the A375 and PC9 signalome cell lines with hundreds of characterized chemical perturbants. Altogether, our results suggest an explanation as to how a chemically diverse collection of drugs converged onto a much smaller number of signaling states. Growth and division are fundamental processes that are subject to multiple mechanisms of homeostasis. While different drugs affect different intracellular mechanisms, an influence on growth or division is a common denominator of many different drug targets. According to this model, the question of whether a drug promotes the p38- or p53-signaling states is not answered by the affected drug targets but rather, by how that drug targets relates to growth and division, i.e., to cell size (Fig. 6H).

The association of p38 with changes in cell size is consistent with previous reports[25,41] whereby p38 MAPK was shown to selectively activate cells that are smaller than their target size. With the present study, we not only confirm the selective association of p38 activity with decreased cell size but further demonstrate that this small-cell-size-dependent signaling is not exclusive to p38 MAPK, as it is also observed with ERK, JNK, AKT, and PKA which together constitute the "p38-signaling state". In addition, our findings also suggest large-cell-size-dependent signaling by members of the p53-signaling cluster. In the context of cell-size regulation, these findings on p53 may prove pivotal. While literature on cell-size checkpoints address mechanisms that are activated in inappropriately small cells, literature on mechanisms that are activated in cells that are larger than appropriate are sparse. Our findings on large-cell-size-dependent p53 activity was confirmed in both cancer cell lines as well as non-cancer cell lines. Both the association of p53 with cell-size homeostasis and the negative correlation of p53 and p38 activity are consistent with previous literature. In previous studies, a negative feedback circuitry linking p38 and p53 has been characterized[42,43]. Specifically, these studies have shown that while p38 promotes the activation of p53, active p53 subsequently responds in negative feedback to inhibit p38, bringing about the resumption of homeostasis[42,43]. Our findings on the cell-size-dependent p53/p38 dichotomy is interestingly consistent with the roles of these proteins in cell growth. While the small-cell-size-dependent activation of p38 is consistent with the mTORC1-activation[44,45] and growth-promoting influence of p38, the large-cell-size-dependent activation of p53 is consistent with the mTORC1-inhibiting and growth-inhibiting influence of p53[46,47].

While previous studies have established a p38-dependent cell-size checkpoint in mammalian cells[25], these studies did not focus on the critical need to monitor cell size in continuously proliferating populations. The present work links size sensing in proliferating cells with an adaptation to homeostasis of growth and cell division. It is also interesting to note that, while both p38 and p53 are well-established stress proteins, their physiological response to stress conditions is very distinct. Stress conditions

that activate p38 typically promote inflammatory programs which promote growth and suppress apoptosis[48]. By contrast, the activation of p53 is both pro-apoptotic and functions to suppress mTORC1-mediated biosynthesis[49].

Overall, the visual barcodes are an easy-to-implement system that can help researchers to multiplex cells for a very wide variety of applications. The system is highly modular and can serve to generate Signalome cell lines with different reporters and thus may be useful in the research of a very wide variety of biological fields.

## Methods

**Cell lines and reagents.** Experiments were performed using the BRAF-mutated melanoma A375 (ATCC CRL-1619) and SK-Mel-5 (ATCC HTB-70) cell lines and the non-small-cell lung cancer EGFR-mutated PC9 cell lines. PC9 was a gift from Dr. Channing Yu of the Broad Institute of Harvard and MIT. Additional experiments were performed using HeLa (ATCC, CCL-2) and the retinal pigmented epithelial (RPE1, ATCC, CRL-4000) cell lines. All cell lines were grown in Dulbecco's Modified Eagle's Medium (DMEM) (Invitrogen, #10569-010). Growing media were supplemented with 10% fetal bovine serum (FBS) and 1% penicillin–streptomycin, pyruvate, and glutamine (Invitrogen, #15140-122).

**PCR for detection of mycoplasma.** The protocol is based on the Takara Kit (#6601). The buffer used, deoxynucleotide triphosphates (dNTPs) and Taq polymerase were obtained from the Takara Ex Taq kit (#RR001A). The following forward and reverse Mycoplasma-specific primers were used for PCR: 5′-ACACC ATGGGAGCTGGTAAT-3′, 5′-CTTCWATCGACTTYCAGACCCAAGGCAT-3′. PCR reactions contained 13.9 μl of Invitrogen UltraPure DDW (10977-015), 2 μl of buffer, 1.6 μl of dNTPs, 0.1 μl of TaKaRa Ex Taq™, 0.5 μl of both forward and reverse primers (final concentration of 20 μM), and 2 μl of genomic DNA or 2 μl of CM. CM was collected after an incubation time of three days on cells, at which point the cells had reached at least 80% confluence. Reactions were held at 94 °C for 30 s to denature the DNA, with amplification proceeding for 40 cycles at 94 °C for 30 s, 55 °C for 2 min, and 72 °C for 1 min.

**Visual barcodes and signalome plasmids construction.** All of the plasmids described in this paper to generate nuclear marker, visual barcodes, and live reporters have been submitted to Addgene and are available. We have also submitted the backbones used to create the visual barcodes and reporters to assist the generation of additional visual barcodes or reporter cell lines. The complete list of the plasmids that we have submitted is shown in Table 1.

To generate the plasmids for the visual barcode clones, we used a CMV-Puromycin-F2A construct on the backbone of pLKO.1 containing a multiple cloning site following the F2A. First, the plasmid was linearized using NheI and MluI right after the F2A sequence. Next, we used Gibson assembly (New England Biolabs, Inc. #E2611) to add the fluorescent protein and localization peptide right after the F2A sequence. All plasmid sequences were verified by sanger sequencing.

To generate the transcription response element (TRE) type of reporter plasmids as well as the AKT reporter plasmid, we first created a promoterless-mStrawberry plasmid (TRE backbone plasmid on backbone of pLKO.1) with a SanDI recognition site before the mStrawberry. First, the plasmid was linearized using SanDI. Next, PCR products containing the promoter region of the plasmids from Supplementary Fig. 2a were fused by Gibson assembly into the TRE backbone plasmid to generate the mStrawberry reporter plasmids.

To generate the translocation reporters, we used the KTR reporters created by Regot et al.[4]. Using Gibson assembly, we introduced these reporters to our TRE backbone plasmid by adding CMV promoter driving the expression of the KTRs fused to mStrawberry. GEMININ reporter was constructed in an identical fashion to the KTR reporters.

**Generating visual barcode reporter clones.** We constructed our visual barcode signalome reporter clones in three steps: (1) for visual demarcation of the nuclear region we infected the cancer cell lines with lentiviruses containing an iRFP-H2A plasmid that we generated. We then generated a single-cell-derived parent clone (see below); (2) we infected the parent clone with a lentivirus containing the visual barcode plasmids and selected for positive cells using puromycin; (3) the puromycin positive cells were then infected with a lentiviruses containing a mStrawberry Signalome reporter and positive cells were selected using blasticidin. Next, we derived single-cell clones from the puromycin-blasticidin positive cells and tested the clones for their visual barcode and reporter activity using known activator/inhibitors of the signaling pathway (Supplementary Fig. 2f).

For generating the lentiviruses, plasmids were transfected into the HEK293T (ATCC, CRL-3216) cells-2nd generation lentivirus system using jetPEI (Polyplus transfection) according to the manufacturer's protocol. 12 h post transfection the 293T cells media was replaced with a fresh media containing 30% FBS to increase the 293T cells population doubling. Twenty-four hours later, lentivirus containing supernatant was collected from the 293T cells growing plates and filtered through a

**Table 1 Complete list of plasmids described in this paper to generate nuclear marker, visual barcodes, and live reporters.**

| Addgene ID | Plasmid | Purpose |
|---|---|---|
| 158665 | Barcode backbone | This plasmid serves as the backbone for the visual barcodes described in our paper. It has a CMV-Puro-F2A backbone to which we insert a fluorescent protein cellular localization combination |
| 158666 | mTAGBFP2 WC visual barcode | Tagging clonal populations with whole-cell mTAGBFP2 fluorescent protein |
| 158667 | mTAGBFP2 NES visual barcode | Tagging clonal populations with NES mTAGBFP2 fluorescent protein. Adding NES to mTAGBFP2 WC |
| 158668 | mTAGBFP2 Peroxisome visual barcode | Tagging clonal populations with Peroxisomal mTAGBFP2 fluorescent protein. Adding Peroxisome localization signal to mTAGBFP2 WC |
| 158669 | mTurquoise2 WC visual barcode | Tagging clonal populations with whole-cell mTurquoise2 fluorescent protein |
| 158670 | mTurquoise2 NES visual barcode | Tagging clonal populations with NES mTurquoise2 fluorescent protein. |
| 158671 | mTurquoise2 Peroxisome visual barcode | Tagging clonal populations with Peroxisomal mTurquoise2 fluorescent protein. |
| 158672 | acGFP WC visual barcode | Tagging clonal populations with whole-cell acGFP fluorescent protein |
| 158673 | acGFP NES visual barcode | Tagging clonal populations with NES acGFP fluorescent protein. Adding NES to acGFP WC |
| 158674 | acGFP Peroxisome visual barcode | Tagging clonal populations with Peroxisomal acGFP fluorescent protein. Adding Peroxisome localization signal to acGFP WC |
| 158675 | EYFP WC visual barcode | Tagging clonal populations with whole-cell EYFP fluorescent protein |
| 158676 | EYFP NES visual barcode | Tagging clonal populations with NES EYFP fluorescent protein. Adding NES to EYFP WC |
| 158677 | EYFP Peroxisome visual barcode | Tagging clonal populations with Peroxisomal EYFP fluorescent protein. Adding Peroxisome localization signal to EYFP WC |
| 158678 | TRE reporter backbone | This plasmid serves as the backbone for the TRE reporters described in our paper. It's a promoterless-mStrawberry-pGK-BSD backbone to which we insert a pathway-specific promoter before the mStrawberry |
| 158679 | WNT-TRE-mStrawberry reporter | This plasmid is a WNT pathway reporter. It has a 7xTcf promoter driving the expression of mStrawberry-pGK-BSD |
| 158680 | NFKB-TRE-mStrawberry reporter | This plasmid is a NFKB pathway reporter. It has a 3X-KB-L promoter driving the expression of mStrawberry-pGK-BSD |
| 158681 | HIF-TRE-mStrawberry reporter | This plasmid is a HIF1A pathway reporter. It has a 6x HIF binding element promoter driving the expression of mStrawberry-pGK-BSD |
| 158682 | YAP/TAZ-TRE-mStrawberry reporter | This plasmid is a YAP/TAZ pathway reporter. It has a YAP/TAZ-responsive synthetic promoter driving the expression of mStrawberry-pGK-BSD |
| 158683 | RAR-TRE-mStrawberry reporter | This plasmid is a RAR pathway reporter. It has a Retinoic Acid Receptor Response Element promoter driving the expression of mStrawberry-pGK-BSD |
| 158684 | p53-TRE-mStrawberry reporter | This plasmid is a p53 pathway reporter. It has two copies of wild-type p53 binding sites promoter driving the expression of mStrawberry-pGK-BSD |
| 158685 | AKT-translocation-mStrawberry reporter | This plasmid is an AKT pathway reporter. It has 1EF1a promoter driving the expression of truncated FoxO1 fused to mStrawberry-pGK-BSD |
| 158686 | ERK-KTR-mStrawberry reporter | This plasmid is a ERK pathway reporter. It has CMV promoter driving the expression of ERK-KTR fused to mStrawberry-pGK-BSD |
| 158687 | p38-KTR-mStrawberry reporter | This plasmid is a p38 pathway reporter. It has CMV promoter driving the expression of p38-KTR fused to mStrawberry-pGK-BSD |
| 158688 | PKA-KTR-mStrawberry reporter | This plasmid is a PKA pathway reporter. It has CMV promoter driving the expression of PKA-KTR fused to mStrawberry-pGK-BSD |
| 158689 | JNK-KTR-mStrawberry reporter | This plasmid is a JNK pathway reporter. It has CMV promoter driving the expression of JNK-KTR fused to mStrawberry-pGK-BSD |
| 158690 | Geminin-mStrawberry reporter | This plasmid is a Geminin cell cycle reporter. It has CMV promoter driving the expression of Geminin fused to mStrawberry-pGK-BSD |
| 158691 | iRFP-H2A | This plasmid encodes an iRFP fused to H2A histone nuclear marker. It has a CMV promoter driving the expression of iRFP fused to H2A. It has no selection markers |

All plasmids listed have been submitted to Addgene.

0.45 µM filter. Next, we removed the target cells (A375, PC9, SK-Mel-5) media and replaced it with the virus-containing filtered media for a period of 24 h. Finally, the virus-containing media was washed and the cells were subjected to positive selection as detailed above.

**Generation of single-cell-derived clones**. Cells (0.5 cells/well in 150 µl) were seeded on a Corning 96-well plate (Cat. Number 3595). After 6–8 h, wells were manually screened for the existence of a single cell in each well. Wells with more than one cell were excluded from further handling. After 2–3 weeks, clones were propagated to bigger wells to generate cell lines that were validated for their reporter activity as indicated in Supplementary Fig. 2f.

**Mice experiments using visual barcodes**. Nine subclones (BFF, CFP, and GFP in WC, NES, and peroxisome) were mixed in equal proportions and injected (2.5 million cells in total) into the flanks of female nude mice (Harlan, Israel). Following 4 weeks of the growth, the mice were sacrificed and the tumors were extracted.

The extracted tumors were broken down into a single-cell suspension using the cold protease method described by Adam et al.[50] (PMID: 28851704). In short: Tumors were incubated at 6 °C for 7 min in a dissociation buffer containing Bacillus Licheniformis protease (10 mg/ml final concentration), PBS and DNaseI(125U/ml). Next, the tumors were transferred to GentleMACS C-tubes (miltenyibiotec) and placed in the gentleMACS Dissociator (brain_03 program, miltenyibiotec). Following dissociation, the cells were sequentially filtered on 70- and 40-µm strainers and spun down at 500xg for 5 min at 4 °C and resuspended in 50 µL cold PBS.

Mice studies were approved by the institutional animal care and use committee of the Weizmann Institute (00400120-3).

**ImageStream analysis**. Cells were imaged by an Imaging Flow Cytometer (ImageStreamX Mark II, AMNIS corp. - part of Luminex, TX, USA). Data were acquired using a ×60 lens, and lasers used were 405 nm (30 mW), 488 nm (30 mW), 561 nm (200 mW), 642 mW (150), and 785 nm (5 mW). Data were analyzed using the manufacturer's software IDEAS 6.2 (AMNIS corp.). Images

were compensated for spectral overlap using single stained controls. Cells were first gated according to their area (in μm²) and aspect ratio (the Minor Axis divided by the Major Axis of the best-fit ellipse) of the iRFP staining. Cells were further gated for focus using the Gradient RMS and contrast features (measures the sharpness quality of an image by detecting large changes of pixel values in the image). Cropped cells were eliminated using the bright-field Area and Centroid X (the number of pixels in the horizontal axis from the upper, left corner of the image to the center of the mask) features. Cells were divided to CFP+, BFP+ and GFP+ according to their corresponding intensities. To identify the three cell morphologies, two features were calculated for each of the FPs used: area of the highest intensity pixels using the Threshold mask, and the similarity feature (a measure of the degree to which two images are linearly correlated, calculated as log-transformed Pearson's Correlation Coefficient) calculated between each staining and the iRFP signal. Plotting these features on a bi-variate plot gave a clear distinction of the three morphologies.

**Drug libraries.** Three drug libraries were used in the screen. The first and second libraries contained 75 and 247 drugs respectively (Supplementary Data 1) that were selected from the Selleck chemicals bioactive screening libraries and purchased from the G-INCPM at the Weizmann Institute (https://g-incpm.weizmann.ac.il/units/WohlDrugDiscovery/chemical-libraries). The third library was a gift from Pfizer and contained 175 drugs (Supplementary Data 2). All libraries were screened at a final concentration of 0.5 μM.

**Screening and imaging.** For screens, clones were grown in 15-cm culture plates (Thermo Scientific, #168381) overnight in DMEM supplemented as above. Before seeding, cells were detached by trypsin (Trypsin EDTA Solution A (0.25%), EDTA (0.02%), 03–050–1B, Biological Industries) and resuspended in imaging media (DMEM without phenol red (01–053–1A, Biological Industries), supplemented as above). The number of cells/ml was counted by Vi-cell XR (Beckman Coulter) and all clones were brought to the same cellular concentration. Clones were then mixed in equal proportions and seeded using EL406 washer dispenser (BioTek) at 5000 or 2500 cells/well onto clear bottom 96 or 384-well plates, respectively (Greiner, product #60–655090, Greiner, product #781-091). Cells were seeded in 135 μl/well in 96-well plates or in 45 μl/well in 384-well plates. We also seeded each of the clones in a separate well to allow training of the CellProfiler analyst software to identify the different clones (see below). Note that in the screen that included 247 drugs the clone that contained the reporter for geminin was not included in the mix as for a technical mistake and consequently data from this clone was not generated in this screen. Plates were then cultured for 24 h at 37 °C, 5% CO₂, and 100% humidity. After 24 h, cells were imaged with the Operetta CLS™ High Content Imaging System (Perkin Elmer) using 10x high NA objective before treating with 15/5 μl of 10× drugs for 96/384-well plates using the CyBi-Well Vario 96/250 Simultaneous Pipettor (CyBio). Next, the cells were imaged in 12-h intervals for a period of 48–72 h. Temperature (37 °C) and CO₂ (3%) were held constant during the imaging. All images were acquired with the same contrast and brightness parameters controlled by the Harmony software (Perkin Elmer). In all, 4–9 fields were acquired from each well in a 384-well plate using a ×10 objective in both digital phase-contrast (DPC) as well as in each of the six FP-specific wavelengths.

**Image analysis and feature extraction workflow.** We first used CellProfiler (Version 2.2.0) to detect nuclei (iRFP), segment cells (DPC), identify tertiary objects (cytoplasm, perinuclear, cell-specific background), and detect and quantify fluorescent proteins of visual barcodes (BFP, CFP, GFP, YFP) and reporters (mStrawberry) in all cellular compartments from all images. The resulting data, termed cytological profiles, consist of more than 200 features that describe the characteristics of each cell such as its size, shape, and the intensity and texture of all FPs expressed. Results of this pipeline were exported both to the spreadsheet and sql-lite database.

**Barcode-deconvolution workflow.** To determine the visual barcode identity of each cell, we used CellProfiler Analyst (Version 2.0) classifier supervised machine learning software. We first trained the software with images from our control wells that were plated with only one subclone type per well (one visual barcode). We made sure that for each clone we trained on at least 200 cells, representing all time points in the experiment. We instructed the CellProfiler to use 50 rules in order to differentiate between the clones. This type of training was done for each of our experiments as we noticed that using the same set of rules between experiments reduces the overall accuracy of barcode calling. Note that a feature could be used more than once in the classifier. Finally, we applied the rules to all cells in the experiment in order to determine the barcode of each cell.

**Data analysis.** Data analysis and statistical tests were performed using R (R version 3.6.0) and RStudio (Version 1.2.5033). Briefly, a metadata file containing each well treatment is joined to the experiment raw data (per cell table with cell features and predicted barcode and reporter). Next, for each cell, we calculated the mStrawberry (the FP used for our reporters) cytoplasmic to nuclei ratio and the mean intensity of the nuclei after subtracting the cell-specific background intensity.

Next, the combined data frame is grouped by time-point, treatment and reporter and Kolmogorov–Smirnov (KS) test is being performed on these two new calculated features. For the translocation reporters the test is being calculated on the treatment's cytoplasm to nucleus ratio density plot and for the transcription reporters, the test is being calculated on the nuclei after subtracting the cell-specific background. The result is an activity score for each reporter at a given treatment and time-point (the score was not calculated in cases where the group contained less than 30 cells). These activity scores were used to perform Hierarchical clustering (distance = Euclidean, agglomeration method = Ward.D2) on the data using the pheatmap package (version 1.0.12). Plotting the activity score was done using the ggplot2 package (version 3.3.0). KS as well as other statistical tests were performed using the stats package (version 3.6.0).

**Reporter activity score.** To measure the effectiveness of treatments, we used a modified KS test, comparing the treated population's intensity distributions with respect to the control (Supplementary Fig. 2d). The test, which measures the biggest difference between the two CDFs (Cumulative Distribution Function) allows us to identify almost every significant effect the treatments had on the reporters' activity. However, since the KS statistic is the absolute value of the maximum difference in CDFs, we assigned a sign to the statistic based on the location of both populations' median values. Thus, our score is ranged between −1 for maximum inactivation, as shown for ERK translocation reporter treated with trametinib (Supplementary Fig. 2d) to 1, maximum activation. In general, we noticed a high correlation between the KS score and the effect of drugs on the distribution mean (data not shown).

**Western blot.** A375 cells were plated a day before treatment on a 10-cm plate at a $1.5 \times 10^6$ cells/well, and were treated by either DMSO or with the following drugs: 1 μM vemurafenib, 5 μM Nutlin 3a, 0.5 μM GSK-269962A, and 0.5 μM GSK2334470, for 48 h. Cells were then lysed with 100 μl of ice-cold RIPA buffer (Thermo Scientific, Piece #89901) on ice. Samples were mixed with 4× protein sample loading buffer (Li-Cor #928–40004) and 10x sample reducing agent (Li-Cor #B0009) and run on a 4–12% Bis-Tris gel at 120 V. Transfer to membranes (Sigma Aldrich Cat# 10401380) was done using Program 2 on the Pierce G2 Fast Blotter (Thermo Fisher Scientific). 1st Antibodies were used to perform immunoblotting, according to antibody manufacturer specifications. Near-infrared (NIR) fluorescence was detected with the Odyssey CLx Infrared Imaging System (Li-Cor biosciences), and signal intensity was quantified with ImageStudioLite software (Li-Cor biosciences). Proteins of interest were normalized to DMSO controls and the GAPDH loading control. Anti-mouse secondary Ab and anti-rabbit secondary Ab were purchased from Li-Cor (#926–32211, #926–68070). The antibodies used in our work are summarized in Table 2.

**RAR/RXR pathway activity.** Three cohorts of melanoma patients that contain expression data were downloaded from the TCGA and the gene expression omnibus (GEO, GSE65904, GSE59455). To calculate pathway activity metrics, we used the PathOlogist tool, which uses gene expression levels and prior knowledge about the interactions within a pathway (as explained by Ben-Hamo et al.[18]).

**Cancer proteome comparisons.** To quantify signaling activities in clinical tumor cell data, we relied on the Cancer Proteome Atlas (https://tcpaportal.org/tcpa/download.html) to retrieve quantitative protein expression data collected from 8167 human tumors that span 32 different types of cancer from the TCGA (https://www.cancer.gov/tcga). To compare signalome reporters with protein expression data from the TCGA, we assembled a list of eight proteins or phosphoproteins that are known to correlate with the activity of pathways included in the signalome (AKTpS473, MAPKpT202Y204, JNKpT183Y185, p38pT180Y182, PKCALPHApS657, NFKBP65pS536, p53, YAP). TCGA protein-level data were correlated to generate the full pairwise-correlation matrix for each type of cancer. Cancers

**Table 2 List of primary antibodies used.**

| Antibody | Company | Catalogue |
|---|---|---|
| pp65 | Cell Signaling | #3033 |
| YAP/TAZ | Cell Signaling | #8418 |
| p-YAP/TAZ | Cell Signaling | #4911 |
| AKT | Cell Signaling | #2920 |
| pAKT | Cell Signaling | #4060 |
| p53 | Cell Signaling | #48818 |
| JNK | Cell Signaling | #9251 |
| p-JNK | Cell Signaling | #3708 |
| pERK1/2 | Cell Signaling | #4370 |
| ERK1/2 | Cell Signaling | #9107 |
| p-CREB | Cell Signaling | #9198 |
| GAPDH | Cell Signaling | #2118 |

with over 30 samples were further analyzed for the similarity of their pattern of correlations between protein levels and the corresponding pattern of correlations observed in the signalome. To do this, the correlation matrices were edited to remove the diagonal elements (as these are trivial). Then, each row of the cancer's correlation matrix was again correlated with the corresponding row of the signalome correlation matrix. This gave a similarity score for each pathway in each cancer. This data was grouped to find how similar the cancers were overall to the signalome in terms of correlations between pathways. Analysis was done with the pandas (version 1.01), numpy (version 1.18.1), matplotlib (version 3.1.3), and seaborn (0.10.0) libraries through the Anaconda Python distribution, and with ggplot (version 3.3.0) through R (version 3.6.2) for visualization.

**Measurements of cell growth.** To measure the average cellular growth rate (protein accumulation rate) in each condition, we used a previously described method for quantification of total macromolecular protein mass in individual cells[28–30]. At intervals during drug treatment, samples were fixed and permeabilized, and cells were reacted with a succinimidyl ester that is covalently bound to a fluorescent dye (SE-A647). SE-A647 covalently binds to fixed proteins to produce a fluorescent signal that is proportional to cell mass as shown in Supplementary Fig. 5A and B. After fixation and staining with SE-A647 (protein) and DAPI (DNA), widefield fluorescence images were collected. The bulk protein content (total SE-A647 intensity of sample) and the number of cells were measured in each sample. From these measurements, we calculated the average growth rate and cell cycle length of cells in each condition, by fitting all data points (from two replicates of each condition) to an exponential growth model. Prior to fixation, throughout the course of drug treatment, proliferation was independently monitored by periodic imaging of live cells via differential phase-contrast imaging. These measurements were used to estimate the average cell cycle length in each condition by fitting data to an exponential proliferation model.

Cells were imaged using a Perkin Elmer Operetta high-content microscope, controlled by Harmony software, with an incubated chamber kept at 37 °C and 5% $CO_2$ during live-cell imaging. A Xenon lamp was used for fluorescence illumination, and a 740 nm LED light source was used for transmitted light. Differential phase-contrast images were collected using a $10 \times 0.4$ NA objective lens. Widefield fluorescence images were collected with a $20 \times 0.75$ NA objective lens.

To slow growth rate the following drug treatments were used: cycloheximide (1, 0.6, and 0.06 μM, Sigma C4859), Torin-2 (10, 5, and 2.5 nM, Tocris 4248), rapamycin (7, 0.7, and 0.07 μM, CalBiochem 553211). To slow the cell cycle, the following drug treatments were used: BN82002 (25, 12.5, 6.2, 3.1, 1.6, and 0.78 μM, Calbiochem 217691), SNS-032 (39 and 9.8 nM, Selleckchem S1145), PHA848125 (175 nM, Selleckchem S2751), Cdk2 Inhibitor III (5, 1.5, 0.75, and 0.38 μM, Calbiochem 238803), Dinaciclib (10, 5, and 2.5 nM, Selleckchem S2768).

**Metric to quantify PCA pre- and post drug treatment.** We developed a metric that quantifies how well measurements of cells that were not exposed to any drug treatment can predict the correlated signaling activities we observed post drug treatment. As shown in Fig. 4, correlated activities in untreated cells seem absent when calculated by pairwise correlations, but are identified by PCA with significance (Fig. 5). The reason for this discrepancy is that pairwise correlations fall short of representing multivariate dependencies. By examining the dataset as separate pairs of pathways, piecemeal comparisons of pairwise correlations reduces the statistical power of any data analysis. On the other hand, performing independent PCA on treated versus untreated also falls short of answering our question. While PCA identified multivariate trends in measurements on both drug-treated and untreated cells, it is hard to say whether such independently identified dependencies are similar.

To circumvent this challenge, we quantified the persistence of the correlations with a different approach. In simple terms, we asked how well principal components calculated from measurements on untreated cells can explain trends that emerge post drug treatment. Intuitively, principal component analysis (PCA) is a method that calculates a new coordinate system that is optimally aligned with linear trends in the measured data (Supplementary Fig. 5B). In the present study, any given drug is scored by measuring its influence on 12 branches of signaling (Supplementary Fig. 5A). In Supplementary Fig. 5A, for example, drugs are represented as points on a 2D coordinate system, where the "coordinates" represent the drug's influence on the measured pathways. In such a case, PCA constructs a new (orthogonal) coordinate system (Supplementary Fig. 5B) that is optimally aligned with linear dependencies in the raw data. Further, the new coordinates calculated by PCA are hierarchically ordered such that the first coordinate (the first principle component) is aligned in a direction that captures the highest variance in the dataset and so on.

To test whether measurements on untreated cells can predict the drug-induced multivariate correlations, we performed a PCA from measurements on cells that were not exposed to drug treatments. This resulted in a coordinate system that we call $\Psi_0$, that has 12 axes in the $\Psi_{0,n}$ directions. Since $\Psi_0$ results from performing PCA on untreated cells, these axes (coordinates) are trivially aligned with correlated signaling pre-drug treatment. What is not clear is whether $\Psi_0$ would also align with the correlations that are promoted by drugs. To test this, we compared the product of the variances of the original measurements to the product of the variance after measurements are projected onto $\Psi_0$.

$$\phi = \frac{\prod_{n=1}^{12} \sigma^2_{\Psi_{0,n}}}{\prod_{p=1}^{12} \sigma^2_p} \tag{1}$$

As is illustrated by Supplementary Fig. 5, the product in the variances describes a region that encloses the data points (in a given coordinate system). When two pathways (here we use the example of p38 and ERK) have activity that's correlated, the region enclosing the data will be smaller in a coordinate system that is aligned with those correlations. A useful aspect of $\phi$ is that it is naturally normalized to the range of (0,1). To begin, if measured activities are not correlated, the volume that encloses the data should not change no matter what coordinate system it is projected onto and the value of $\phi$ will be near one in all coordinates. By contrast, if we imagine perfectly correlated data, $\phi$ would tend to zero (e.g., a line has no volume).

In our work, this was used to see whether the linear combinations of pathway activity seen in unperturbed cells reflect correlations in drug-treated cells better than the pathway activities themselves: low values of $\phi$ indicate that these combined signaling groups do not reflect the coordination of signaling changes in response to drugs, while higher values indicate that they do.

**Qualify drugs conformity to p38- or p53-signaling state.** To score the extent to which a given compound promotes the p38-signaling vs p53-signaling states, we used Eq. 2.

$$\psi = \frac{PC_1}{\sqrt{\sum_{i=1}^{12} PC_i^2}} \tag{2}$$

Here, the value $PC_i$ is the numerical value of the drug's effect in the $i$th axis of the principal coordinate system. Intuitively, the extent by which a drug conforms with the p38- and p53-signaling clusters is represented by the extent to which its influence on the 12 pathways is aligned with the first principal component: the magnitude of which is $PC_1$. Equation 2 is the ratio of a drug's influence on $PC_1$ as compared to the size of the effect vector. This quantifies how closely aligned the drug's effect is with p38–p53 signaling.

**Linear deconvolution of drug screen.** To estimate the quantitative effect of each drug target, we fit a Ridge regression model, optimizing the following loss function:

$$\sum_{i=1}^{n} \left( y_i - \sum_{j=1}^{p} x_{ij} \beta_j \right)^2 + \lambda \sum_{j=1}^{p} v_j \beta_j^2 \tag{3}$$

Where $y$ is the vector of responses, $x$ is the compound target matrix, $\lambda$ is our overall regularization term, and $v_j$ is additional regularization applied only to interaction terms i.e., if a target is not an interaction term $v_i = 1$ otherwise $v_j > 1$. In this paper we are interested in the estimated individual effect for each target, so we only looked at the individual effect estimates after fitting.

We used fivefold cross-validation to select both $\lambda$ and $v$. We searched a grid space of possible $\lambda$ value combinations with $v$, choosing the set with the smallest MSE of all combinations tried. Starting with $v_j = 1$ and doubling it's value each iteration we tried searching up till $v_j = 2^{30}$ where if the model had not found a local minimum by that point we would not include the interaction terms in the model since the smallest error continued to be the model that removed the interaction terms as much as possible, this was the case in our analysis of cell size. The $\lambda$ values were the default selection using library glmnet in R.

**Adherence to PC1 and drug strength.** While hierarchical clustering assigned each drug with a one of three clusters, the extent to which individual compounds conform with their assigned cluster differs from one drug to another. To score the association of individual drugs with clusters A–C, we relied on the scores retrieved from the principal component analysis (PCA). Drugs that promote the p38- or p53-signaling states are scored with positive or negative scores in PC1 (respectively). We therefore calculated two separate metrics. The first,

$$Adherence\ to\ PC_1 = \frac{|PC_1|}{\sum_i (PC_i)^2} \tag{4}$$

is a measure of the extent to which the influence of a given drug is explained by PC1 alone. The second metric,

$$Drug\ Strength = \sqrt{\sum_i (PC_i)^2} \tag{5}$$

is a measure of the total sum of the influence that the compound had on the 12 signalome reports.

**Enrichment strength.** To identify functional associations that characterize the p38- and p53-signaling states, we relied on the STRING database. As seeds for the analysis, we relied on the two lists of drug targets; (1) drug targets that—when inhibited—result in the activation of the p38 signaling state and (2) drug targets that—when inhibited—result in the activation of the p53-signaling state. The third

signaling state (Cluster C) had too few unique targets for statistically significant enrichment.

These two lists of drug targets were used as seeds for the STRING database. STRING is a widely used, constantly updated, and expanding database of PPIs[51], used for the examination of verified, or potential interactions among proteins of interest. These networks are rich in information on protein clusters and functions based on various sources including KEGG[52] and uniprot[53]. The analysis provided by STRING is characterized by two stages of analysis: (1) construction of a genetic association network (B) functional enrichment. To enable functional enrichment scores, STRING links to annotation databases including KEGG and uniprot annotated keywords. For both the p38- and the p53-drug target lists, we performed functional enrichment with both KEGG and uniprot. To score for the different function enrichment categories, we used the statistical strength score provided by STRING, a statistic describing how large the enrichment effect is. Basically, the enrichment strength is the log ratio (The statistical strength is calculated as the log-ratio log10[observed/expected]) whereby, expected is the number of proteins in the p38- or p53-subnetworks that are annotated with a given term and expected is the number of proteins that are annotated with this term in a random network of the same size.

**Reporting summary**. Further information on research design is available in the Nature Research Reporting Summary linked to this article.

## Data availability

All data generated or analyzed during this study are included in this published article and its supplementary information files. The different datasets that were used for RAR//RXR patient stratification are publicly available as follows: (a) Cirenajwis et al. dataset is available in GEO under the accession number: GSE6590; (b) Budden et al. dataset is available in GEO under the accession number: GSE59455; (c) Human melanoma tumors from TCGA were downloaded from the GDC data portal https://portal.gdc.cancer.gov/. All the data relating to the TCGA protein analysis was downloaded from https://tcpaportal.org/. For the Protein–protein interactions network analyzed, we used https://string-db.org/. Source data are provided with this paper.

## Code availability

The code and example datasets are available as supplementary information files.

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

## Acknowledgements

R.S. is funded by the Israel Science Foundation (grant no. 2927/21), the European Research Council (ERC) under the European Union's Horizon 2020 research and innovation program (grant agreement no. 818086), the Fabrikant-Morse Families Research Fund for Humanity, and the Knell Family Center for Microbiology.

## Author contributions

T.K., E.N., V.R., and R.S. conceived and initiated the project and analyzed the data. T.K. and E.N. designed and carried out the experiment. T.K. and N.F. wrote data analysis code. S.I., R.B.H., M.B.G., N.P., J.H., and A.H. contributed to the data analysis. Z.P. contributed to the design and execution of the ImageStream experiment. T.K., E.N., N.P., M.B.G., R.K., and R.S. prepared the manuscript. R.K. and R.S. supervised the project.

## Competing interests

R.S. received a grant from Merck EMD Serono, is a paid adviser to Curesponse, and Baccine and is serving on the SAB of Micronoma. The remaining authors declare no competing interests.
