## [Peer Review File · Nature Communications]

Visual barcodes for clonal-multiplexing of live microscopy-based assaysReviewers' comments:

Reviewer #1 (Remarks to the Author): with expertise in imaging single-cell network, reporters.

Kaufman et al. create a visual barcoding system using three localization markers and four fluorescent proteins. Using this ability to differentiate 12 cell lines in one imaging experiment, the authors are capable of monitoring the activity of 12 different signaling pathways simultaneously in one well of an imaging plate. As a proof of principle, the authors use this system in combination with drug libraries and high throughput imaging to find that most compounds behave in one of 2 ways: increasing ERK, JNK, p38 and AKT and decreasing WNT, p53, NFkB, RAR, HIF and YAP/TAZ or vice versa. The authors conclude that this bimodal behavior depends on the effects of drugs on growth or proliferation. The paper is well written and the experiments and analyses are technically sound. I find the localization based barcoding strategy clever and of interest. However I have major concerns about potential artifacts in the way the study is designed.

1- I see as the major novelty of the study the barcoding. However I don't think the authors thoroughly explore the limits of this strategy. For example nuclear FPs are distinguishable from their set of 3 (WC, Peroxi, NES) and could add a significant amount of combinations. Using tandem FPs could also increase the throughput significantly.

2- Surprisingly all pathways that are measured by using cytoplasmic to nuclear intensity ratios fall into cluster A and all pathways monitored using transcriptional readouts fall into Cluster B. Is the correlation a result of the reporter type rather than biological? In fact I find it surprising that the 2 modules identified go together. Other studies have seen inverse correlations between p38 and ERK activity for instance. The authors should validate the results using other reporter types.

3- In Fig 6, where authors use protein biosynthesis inhibitors (such as CHX) or rapamycin to suggest that cluster B is inhibited by "cell growth". However these drugs may affect overall transcription or translation rates compromising the sensor readout. Controls with destabilized gene expression reporters that do not depend on signaling should be included.

4- I found title and abstract a little misleading and was expecting to see that authors had managed to add 12 reporters into a single, monoclonal cell line rather than barcode many different clones in a single imaging well. Minor rewording in the title/abstract would be recommendation to manage expectations.

Minor Concerns

1- Pics in fig1D YFP pannel look switched.

2- Authors mention "cluster C" which does not follow this p38-p53 dichotomy, but do not comment further on this. Some more analysis on drug types/experiments may yield interesting results as to why this third population exists, and if these drug classes are capable of breaking an important network level regulation. What is different about Cluster C?

3- Clusters A and B suggest a coordinated signaling response at the network level. If they cluster their drugs that give these two outputs based on characterizations such as targets, would they be able to hypothesize a mechanism for this separation?

4- Authors dichotomize p38 and p53, showing opposing effects but fail to put this in context with the literature that suggest p38 promotes p53 expression.

Reviewer #2 (Remarks to the Author): single-cell quantitative imaging, therapy resistance, drug screening, systems biology

The authors describe the development of an intriguing approach to allow the simultaneous readout of multiple signaling pathway from the same cell population. This is achieved by generating clonal cell lines that can be distinguished based on "cellular barcodes" consisting of fluorescently labeled cell compartments. Each uniquely labeled cell line is then engineered to report on one of 12 broadly relevant cancer signaling pathways. The resulting, uniquely labeled reporter cell lines are mixed to generate a cell population that enables the deconvolution of 12 pathway readouts using fluorescence microscopy. Using this mixed reporter cell population, the authors monitored the response to 422

chemical perturbations and identified two anti-correlated signaling clusters that appear to be related to growth homeostasis.

While the study shows a creative approach to maximize the signaling information acquired from the same cell population, the useful applications of the barcoding strategy described here are not obvious and are not demonstrated in the text. Notably, a significant limitation of the proposed technology is that it doesn't capture truly multiplexed information on a single cell level, as the readouts of the individual pathways originate from different cell sub-populations. Furthermore, the biological findings in the authors' biological example are too superficial to bring clarity to the important problem of regulation of cell size.

Below we offer some suggestions in the hopes that it will be useful to the authors.

Major Concerns:

1. Given the effort of engineering the 9 or 12 reporter cell lines, a conventional experiment with the same markers in an arrayed format or grouped into 2-3 marker sets (with the markers spread out across 4 or 5 optical channels) would be a preferable alternative for the shown experimental scale (422 drug screen). The conventional experimental setting allows for 4-5 simultaneous pathway readouts on the same cell and would also circumvent concerns about the effects of low and/or uneven number of cells per signaling readout.
2. The key value added of live cell imaging is the ability to track the behavior of individual cells over time. Unfortunately, the proposed approach of multiplexing cell populations only allows for the limited extrapolation of the signaling states across different cells. Such an extrapolation is challenging because cancer cells are notoriously heterogenous, which hinders interpretation of the results. Given that the authors do not use the live cell information, it would seem that fixed cell-level multiplexing approaches (e.g. CODEX (Goltsev et al Cell 2018) or 4i (Gut et al Science 2018)) would accomplish the same task at least as well.
3. Interpretation of the Komolgorov-Smirnov (KS)-based activity scores is difficult because these fluorescent markers are not benchmarked against gold standard methods (e.g. target phosphorylation by Western blot) of monitoring pathway outputs. More experiments are needed that demonstrate the relationship between the activity score and changes in pathway output, as well as controls that demonstrate that expressing any individual reporter does not change the output of any of the pathways being measured.
4. The interpretation of Figures 3-6 hinges on the concept of two distinct clusters (dubbed p53 and p38) in the cell population. This analysis is purely correlative and leaves far more questions and concerns open than it provides answers.
 - First, the focus on p53 and p38 over the other 11 markers studied is unsubstantiated. The importance of either p53 or p38 for the observed signaling states is unclear.
 - Second, while a regime of cells with high p53 and low p38 is visible in Figure 3B, a low p53 and high p38 cluster is not resolved as distinct from the non-bioactive drugs.
 - Third, a Cluster C, which is at least twice the size of the extensively discussed Cluster A, does not show inverse correlation between p53 and p38 and is omitted from all further discussion without explanation.
 - Fourth, the authors make no attempt to analyze how the diversity in composition of the drug library affects clustering. Do drugs with similar mechanisms of action fall into the same cluster? Does cell cycle phase play a role in the clustering?
 - Finally, the authors go on to state that the separation between p53 and p38 states exists in untreated cells based on "noise" in the experiment. This interpretation is unconvincing and not supported by data. An alternative explanation could be that the observed clusters are simply the result of mixing separately engineered clonal cell populations.
5. While we recognize the value and interest in the long-standing question of regulation of cell size, the analysis to this effect is very unsatisfying and completely lacks any mechanistic insight. A more detailed investigation of the biological mechanisms regulating the observed anti-correlative signaling clusters would significantly strengthen the study.

Minor Concerns:

Various aspects of the paper appear hastily constructed and lack sufficient attention to detail. We include a non-exhaustive list of examples below.

1. The omission of Geminin data beginning in Figure 3 due to “technical error” is concerning, especially because the authors go on to discuss the balance between division and cell growth in detail. It would seem a cell cycle marker would be particularly useful for the authors’ later analyses.
2. Justification or experimental detail is lacking from various figures. For example, no justification is given for the selection of a bioactivity cutoff of 0.2 in the authors’ activity score; no sample sizes are offered in any of the data in Figure 2; the data in Figure 3 use only the 122 bioactive drugs, whereas the data in Figure 4 use the full set of 422 drugs; some images in Figure 1D appear to be scrambled.
3. Many of the plots throughout the paper are difficult to interpret because of the presence of hundreds of overlapping points. A data presentation with a heatmap of point density would be substantially more informative.
4. Figure 6B abruptly changes cell lines without clear justification.

We would like to thank the reviewers very much for all of the insightful comments. We have addressed all of them as requested and feel that this work has contributed significantly to the clarity and rigorosity of our manuscript. Below please find a detailed point-by-point response to each of the specific points that were raised by the reviewers.

Comments from Reviewer #1 (Remarks to the Author): with expertise in imaging single-cell network, reporters.

1. Reviewer Comment:

Kaufman et al. create a visual barcoding system using three localization markers and four fluorescent proteins. Using this ability to differentiate 12 cell lines in one imaging experiment, the authors are capable of monitoring the activity of 12 different signaling pathways simultaneously in one well of an imaging plate. As a proof of principle, the authors use this system in combination with drug libraries and high throughput imaging to find that most compounds behave in one of 2 ways: increasing ERK, JNK, p38 and AKT and decreasing WNT, p53, NFkB, RAR, HIF and YAP/TAZ or vice versa. The authors conclude that this bimodal behavior depends on the effects of drugs on growth or proliferation. The paper is well written and the experiments and analyses are technically sound. I find the localization based barcoding strategy clever and of interest. However I have major concerns about potential artifacts in the way the study is designed.

1- I see as the major novelty of the study the barcoding. However I don't think the authors thoroughly explore the limits of this strategy. For example nuclear FPs are distinguishable from their set of 3 (WC, Peroxi, NES) and could add a significant amount of combinations. Using tandem FPs could also increase the throughput significantly.

1.1. Author Response:

We agree with the reviewer that a yet higher potential (or higher 'limit') can be gained from our technology if nuclear or other localizations are added or if tandem barcodes are added. In agreement with the reviewer's suggestion, figure #1 in our manuscript demonstrates a five-localization barcode system, rather than the three barcode-system used throughout the rest of the paper and clearly shows a very high precision of recall rate (Fig 1C). It is true that in the latter sections of the study we relied on three, rather than five barcodes, but this is only because a 3-barcode system was found sufficient to discriminate 12 reporters.

To further follow the reviewer's suggestion, in the revised manuscript, we now added a new experiment demonstrating that the complexity of the barcoding can be substantially increased by combining two barcodes in the same cell. Indeed we demonstrate that we can not only combine two barcodes with different colors and localizations but can also combine two barcodes with the same color but different localizations or two barcodes with the same localization but different colors. In all of these combinations a very high precision and recall

rates were observed as now clearly explained in the main text and presented in Sup figs 1D,E. Note that combining two barcodes (composed of 4 colors and five localizations) into a single clone can generate up to 160 different barcodes.

2. Reviewer Comment:

2- Surprisingly all pathways that are measured by using cytoplasmic to nuclear intensity ratios fall into cluster A and all pathways monitored using transcriptional readouts fall into Cluster B. Is the correlation a result of the reporter type rather than biological? In fact I find it surprising that the 2 modules identified go together. Other studies have seen inverse correlations between p38 and ERK activity for instance. The authors should validate the results using other reporter types.

2.1. Author Response:

The reviewer raises the concern that the correlations we observe may derive from artifactual differences in reporter type. Note that all of the reporters that we used have been developed by others and thoroughly validated - as we cite in Sup table 2A. Still - to further validate the correct activity of the reporters in our system we have now taken different approaches to demonstrate the our reporters correctly report on the cellular signaling and that our findings are not a result of a technical issue:

- (1) To validate the accuracy of our reporters we used known activators and inhibitors of all 12 reporters. As can be seen in Sup figure 2F, the reporters correctly detected the expected drug effect in 100% of the cases.
- (2) To further test the accuracy of our reporters using a different method, we now also compared the reporters signaling to protein activity markers as measured by western blot analysis. As can be seen in the new Sup Figure 2G, results of western blot analysis were identical to the results of the signalome reporters, supporting again the accuracy of our reporters.
- (3) Lastely, we have generated Signalome cell lines that have ERK TRE and KTR reporters in both A375 and PC9 cell lines as well as AKT TRE and KTR reporters in the PC9 cell line. While we did not have the drug plates of our big screen any more we tested the effect of specific drugs on these signalome cell lines and observed a highly similar reporting of both the TRE and KTR reporters. Here are a few examples that demonstrate this high correlation:

Cell line	Drug	ERK-TRE	ERK-KTR	AKT-TRE	AKT-KTR
PC9	Gefitinib (EGFRi)	-0.44	-0.66	0.26	0.36

PC9	Selumetinib (MEKi)	-0.08	-0.35	0.14	0.28
A375	Vemurafenib (BRAFi)	-0.21	-0.72	N/A	N/A
A375	Trametinib (MEKi)	-0.28	-0.83	N/A	N/A
A375	MK2206 (AKTi)	0.044	0.2	N/A	N/A
A375	GSK26962A (ROCKi)	0.09	0.08	N/A	N/A

** Number represent the KS-scores after 48 hours of drug treatment.

3. Reviewer Comment:

In Fig 6, where authors use protein biosynthesis inhibitors (such as CHX) or rapamycin to suggest that cluster B is inhibited by “cell growth”. However these drugs may affect overall transcription or translation rates compromising the sensor readout. Controls with destabilized gene expression reporters that do not depend on signaling should be included.

3.1. Author Response:

We appreciate the reviewer’s concern. As the reviewers point out, drugs that lower rates of protein synthesis (e.g. cycloheximide, rapamycin and torin) can artifactually lower levels of expressed reporters – raising concern for potential artifact. We believe that there are multiple lines of evidence to support the fact that our observations are not the result of such an artifact. Here are a few that support our claim:

- A. Measurements in Fig. 6B and 6C panels were not generated by the reporters but direct measurements in other methods. These results are thus completely unrelated to the potential artifact that was raised by the reviewer.
- B. While the reviewer’s concern can explain diminished reporter signals in conditions that inhibit protein synthesis, these concerns cannot explain the converse – increased reporter activity in conditions of inhibited protein synthesis. In our case, it is exactly this ‘non-trivial’ trend that is observed. Drugs that lower rates of protein synthesis result in cluster B type of signaling in which the expression of our transcription-based reporters are increased. To further stress this point, the following figure clearly demonstrates that mTOR inhibitors indeed increase the signal from NFkB (Cluster B pathway). As a control we also added known NFkB inhibitors that

indeed demonstrate inhibition of the pathway.

4. Reviewer Comment:

I found title and abstract a little misleading and was expecting to see that authors had managed to add 12 reporters into a single, monoclonal cell line rather than barcode many different clones in a single imaging well. Minor rewording in the title/abstract would be recommendation to manage expectations.

4.1. Author Response:

We thank the reviewer for this remark. We have now stressed in abstract that different clones of reporters were multiplexed to prevent any misleading conception.

Minor Concerns

5. Reviewer Comment:

Pics in fig1D YFP panel look switched.

5.1. Author Response:

We are sorry for this typo which seems to have been introduced by our graphic editor. This is now corrected.

6. Reviewer Comment:

Authors mention “cluster C” which does not follow this p38-p53 dichotomy, but do not comment further on this. Some more analysis on drug types/experiments may yield interesting results as to why this third population exists, and if these drug classes are capable of breaking an important network level regulation. What is different about Cluster C?

6.1. Author Response:

Following the reviewers suggestion, we attempted to gain insight into the mechanisms that render drugs in cluster C by examining the cluster C - drug targets. Overall, we found these drugs to be diverse and not confined to a single mechanism, as suggested by the lack of coherence that is presented in cluster C (Figure 3A). However, one of the the top hits in cluster C is a CDK4 inhibitor, which we have previously shown (PMID: 34022133) as regulator of cell size homeostasis. This raises the possibility that cluster c represents mechanisms that disrupt cell size homeostasis - a hypothesis which we shall pursue in future investigations and is now mentioned at the very end of the results section.

7. Reviewer Comment:

Clusters A and B suggest a coordinated signaling response at the network level. If they cluster their drugs that give these two outputs based on characterizations such as targets, would they be able to hypothesize a mechanism for this separation?

7.1. Author Response:

Following the reviewer's suggestion, we analysed the drug targets associated with clusters A, B and C for mechanisms that may explain the binary partitioning of the pathways reported by the signalome. As now clearly described in the results section, these analyses strengthened our conclusions by further implicating the binary partitioning of the signalome pathways with cell size control.

Here are a few examples for the new data we added to better establish the connection between the two signaling states and regulation of cell size::

1. In figure 6D we now clearly show a correlation between cell size and the p53 vs p38 signaling states as reflected by the PC1 values of the signalome data.
2. We strengthened the above result by taking data generated by Liu et al. In this study cell size measurements were taken in HeLa cells in response to treatment with a drug library. We show that drugs that inhibited cell growth in Liu et al were identified in our study as targets that promote the activation of the p38 signaling state. The inverse was true for the p53 signaling state. Targets which slowed the rate of division in Liu et al were identified in our study as targets that promote the activation of the p53-signaling state (Figure 6E). Altogether, these results support the universality of our findings beyond our specific signalome cell lines.
3. Further support for the coordinated signaling response depicted by the p53- and p38-signaling states comes from an analysis done using the functional enrichment analysis tool of the STRING database. In this analysis we now show that drug targets that selectively promote the p38- or the p53- signaling states are functionally distinct. We found that drug targets that promoted the p53-signaling led to enrichment in functions relating to cell cycle and cell division (Sup figure 6A). In contrast, drug targets that promoted the p38-signaling state led to an enrichment in anabolic activity functions such as: glucose and carbohydrate metabolism (Sup figure 6B) thus giving support to the connection of those two clusters with cell division and cell growth.
4. Additional support for our model comes from the many experiments that were already included in the original manuscript and can be found in panels: 6B, 6C, 6F, and 6G.
5. We also elaborated in the discussion section on our findings describing the two signaling states and putting them in the context of the broad scientific literature.

8. Reviewer Comment:

Authors dichotomize p38 and p53, showing opposing effects but fail to put this in context with the literature that suggests p38 promotes p53 expression.

8.1. Author Response:

We thank the reviewer for this critique. To the revised manuscript, we now added a section that specifically shows how our findings on p38 and p53 are mainly consistent with the literature (e.g. PMID: 29751559, PMID: 19920204). As noted by the reviewer, p38 is known to activate p53. However, p53 is further known to inhibit p38, resulting in a negative feedback loop that is consistent with the negative correlation of p53 and p38 that is observed in our data. Here is an example from one of the works (PMID: 29751559) describing this negative feedback loop:

Reviewer #2 (Remarks to the Author):single-cell quantitative imaging, therapy resistance, drug screening, systems biology

9. Reviewer Comment:

The authors describe the development of an intriguing approach to allow the simultaneous readout of multiple signaling pathways from the same cell population. This is achieved by generating clonal cell lines that can be distinguished based on “cellular barcodes” consisting of fluorescently labeled cell compartments. Each uniquely labeled cell line is then engineered to report on one of 12 broadly relevant cancer signaling pathways. The resulting, uniquely labeled reporter cell lines are mixed to generate a cell population that enables the deconvolution of 12 pathway readouts using fluorescence microscopy. Using this mixed reporter cell population, the authors monitored the response to 422 chemical perturbations and identified two anti-correlated signaling clusters that appear to be related to growth homeostasis.

While the study shows a creative approach to maximize the signaling information acquired from the same cell population, the useful applications of the barcoding strategy described here are not obvious and are not demonstrated in the text.

9.1. Author Response:

We thank the reviewer for appreciating our creative approach. As for the advantages and possible uses of this technology, the manuscript now puts forward a rather extended list of potential applications that we either demonstrate directly or discuss. To name only a few examples:

- Efficiency - we clearly demonstrated the multiplexing efficiency in multiple systems. One example is the use of one 384-well plate to test the effect of 75 drugs (in triplicates) on 12 signaling pathways over multiple time points in a single-cell resolution.

- Reducing technical noise - while well-to-well variations are always a concern in high-throughput screens, the current system compares all signaling pathways activity in the same well and thus, when correlations between signaling pathways are studied, the source of noise is greatly reduced.
- Applicability to cells in suspension and to in-vivo experiment - our demonstration that image-stream technology can also distinguish the different barcodes open the possibility to use the technology for both cells in suspension and for in vivo experiments as we demonstrate in figure 1.
- Some of the experiments that we did can not be done without measuring multiple reporters in the same well as they depend on variations between wells. See for example figure 5C.
- In the paper we also discuss the potential possibility to use the multiplexed visual barcodes to help with live cell tracking applications as mistakes in the tracking (switching between cells) will cause a switch in the barcode in most of the cases.
- In the revised version of the manuscript we also demonstrate the feasibility of combining two barcodes into one cell thus increasing the potential number of barcodes from ~20 to ~160. Our labs are already using this large number of barcodes for new projects.

10. Reviewer Comment:

Notably, a significant limitation of the proposed technology is that it doesn't capture truly multiplexed information on a single cell level, as the readouts of the individual pathways originate from different cell sub-populations.

10.1. Author Response:

The reviewer correctly identifies an important tradeoff. While having multiple FP reporters in a single cell provides obvious advantages, these advantages come with a drawback; the number of FP that can be expressed in one cell is limited to 2-5 FP. In the revised manuscript we not only demonstrate the ability to multiplex 12 reporters, but also show how this multiplexing can be scaled up to ~160 visual barcodes (Sup figure 1D,E). It should be noted that potentially one can combine 2-3 reporters in the same cells and add a visual barcode to each clone of 2-3 reporters.

11. Reviewer Comment:

Furthermore, the biological findings in the authors' biological example are too superficial to bring clarity to the important problem of regulation of cell size.

11.1. Author Response:

In the revised manuscript, we include new data and analyses linking our results with literature on cell size. Please see **section 7.1.** above for a detailed description.

12. Reviewer Comment:

Given the effort of engineering the 9 or 12 reporter cell lines, a conventional experiment with the same markers in an arrayed format or grouped into 2-3 marker sets (with the markers spread out across 4 or 5 optical channels) would be a preferable alternative for the shown experimental scale (422 drug screen). The conventional experimental setting allows for 4-5 simultaneous pathway readouts on the same cell and would also circumvent concerns about the effects of low and/or uneven number of cells per signaling readout.

12.1. Author Response:

As for the concern regarding uneven number of cells, our results demonstrate that the difference in the number of cells per clone in each well is generally not greater than two fold. As we count the cells before plating and as the proliferation rate of the clones is very similar (Sup figure 2B) we never experienced differences that became a problem. As per the comparison with arrayed reporters or a few FP expressed in the same cell - while these are indeed other screening options, the current technology allows scaleup that can greatly improve the throughput of screens - as we now routinely do in our labs.

Lastly, as all of the measurements that we do are based on averaging signals from single-cells, differences in the number of cells have minimal effect on the median activity score. As expected, the standard errors become larger as the total number of cells/groups drops, and from that reason we always demand to have at least 30 cells/well reporting for each pathway. This limitation is now clearly indicated in the methods section. Please also find below an example in which we randomly selected a well in our experiment that was treated with a ROCK inhibitor and sub-sampled the number of cells that reported for ERK activity down to only 5. As can be seen in the bar plot the KS score demonstrates activation of the ERK pathway in response to ROCK inhibition even if a very small number of cells were selected. The STDEV is becoming larger as the number of cells drops.

13. Reviewer Comment:

The key value added of live cell imaging is the ability to track the behavior of individual cells over time. Unfortunately, the proposed approach of multiplexing cell populations only allows for the limited extrapolation of the signaling states across different cells. Such an extrapolation is challenging because cancer cells are notoriously heterogenous, which hinders interpretation of the results. Given that the authors do not use the live cell information, it would seem that fixed cell-level multiplexing approaches (e.g. CODEX (Goltsev et al Cell 2018) or 4i (Gut et al Science 2018)) would accomplish the same task at least as well.

13.1. Author Response:

We agree with the reviewer that tracking of single cells over time is possible using our system. Indeed, the visual barcodes will greatly contribute to the reduction of tracking errors that are not very rare in these live cell tracking applications. We indeed use such data in a different project that is beyond the scope of this MS. This point is now also brought forward in the discussion section. Note that unlike the systems that the reviewer mentioned, in the experiments done for this MS our dynamic/live approach detects the heterogeneous effects of perturbations over time as we measure the activity of the 12 reporters from the same wells in multiple time points. Using fixed cells would have imposed a significant limit on the potential throughput of the experiment.

14. Reviewer Comment:

Interpretation of the Komolgorov-Smirnov (KS)-based activity scores is difficult because these fluorescent markers are not benchmarked against gold standard methods (e.g. target phosphorylation by Western blot) of monitoring pathway outputs. More experiments are needed that demonstrate the relationship between the activity score and changes in pathway output, as well as controls that demonstrate that expressing any individual reporter does not change the output of any of the pathways being measured.

14.1. Author Response:

We agree with the reviewer that the interpretation of Komolgorov-Smirnov (KS) scores would benefit from a comparison with gold standard methods such as western blotting. This point was also raised by the other reviewers and we have addressed it in a data-driven approach in point #2.1 above.

15. Reviewer Comment:

4. The interpretation of Figures 3-6 hinges on the concept of two distinct clusters (dubbed p53 and p38) in the cell population. This analysis is purely correlative and leaves far more questions and concerns open than it provides answers.

- First, the focus on p53 and p38 over the other 11 markers studied is unsubstantiated. The importance of either p53 or p38 for the observed signaling states is unclear.

15.1. Author Response:

We appreciate the reviewers critique. Putting aside the technological achievement of the barcode system, the central finding in our study is a drug-target independent binary partitioning of signal transduction into two clusters. It is this binary partitioning into clusters A and B that is the claim we make in the paper. As for p53 and p38, we use these as nomenclature to name clusters A and B. We do not make the claim that p38 and p53 are mechanistically responsible for the observed partitioning.

16. Reviewer Comment:

- Second, while a regime of cells with high p53 and low p38 is visible in Figure 3B, a low p53 and high p38 cluster is not resolved as distinct from the non-bioactive drugs.

16.1. Author Response:

We agree with the observation made by the reviewer. The explanation behind this observation is that in the cell line that was used for this figure (A375 melanoma cells), TP53 status is wild type and the basal activity of p53 is very low. In contrast, P38 is very active in these cells, as was also reported by others (e.g. PMID: 31661126). As a result - the

magnitude of potential change by drugs in cluster B is much greater than the magnitude of effects that can potentially be observed in drugs from cluster A.

Our data also support the fact that P53 is not active and P38 is very active in treatment-naive A375 cells. To demonstrate that we plotted the KS scores of both reporters in all of the cells in the DMSO control wells in our screen. The X-axis was normalized using the effect of all 422 drugs by choosing the maximal and minimal KS scores that were reached in this screen. This plot also clearly demonstrates the basal activity of P38 and in-activity of P53 in our cells. (The bimodal nature of the graph is a result of using KS-scores that can almost never have a score of 0).

Note, that while the majority of the drugs elevate P53 and reduce P38 activity, still - there are drugs that drive cluster A signaling as can be seen in the figure.

17. Reviewer Comment:

- Third, a Cluster C, which is at least twice the size of the extensively discussed Cluster A, does not show inverse correlation between p53 and p38 and is omitted from all further discussion without explanation.

17.1. Author Response:

We thank the reviewer for this comment that was also brought up by another reviewer. Please see our answer to 7.1 describing how we addressed it in the revised MS.

18. Reviewer Comment:

- Fourth, the authors make no attempt to analyze how the diversity in composition of the drug library affects clustering. Do drugs with similar mechanisms of action fall into the same cluster? Does cell cycle phase play a role in the clustering?

18.1. Author Response:

We thank the reviewer for this comment. Please see our answer to 7.1 in which we address this comment.

19. Reviewer Comment:

- Finally, the authors go on to state that the separation between p53 and p38 states exists in untreated cells based on “noise” in the experiment. This interpretation is unconvincing and not supported by data. An alternative explanation could be that the observed clusters are simply the result of mixing separately engineered clonal cell populations.

19.1. Author Response:

We thank the reviewer for this comment and would like to take the opportunity to better explain this subtle point. Our data demonstrates a statistically significant separation of the p53 and p38 states in cells that were treated by drugs. The term ‘noise’, however, is confusing and reflects a bad writing style on our end which we have now corrected in the main text. Instead, the separation of the two states in cells that are not drug treated simply suggests that, while the mechanism responsible for separating the two clusters is enhanced by drug treatments, it is not caused by the drug treatments but rather only enhanced by it.

It is very well known that small differences between the wells in multi-well plates may affect cell growth and proliferation as well as many other phenotypes. Indeed, it is a very common practice by many labs not to use the most peripheral rows and columns on multi-well plates as the data of these wells is frequently very different from data coming from the more central wells on the plates. Here we only took advantage of these highly common environmental differences across multi-well plates to demonstrate that the cellular signaling response to these various stresses is still following the two signaling states bifurcation.

20. Reviewer Comment:

5. While we recognize the value and interest in the long-standing question of regulation of cell size, the analysis to this effect is very unsatisfying and completely lacks any mechanistic insight. A more detailed investigation of the biological mechanisms regulating the observed anti-correlative signaling clusters would significantly strengthen the study.

20.1. Author Response:

We thank the reviewer for pointing this out. In the revised manuscript, we now added significantly more data and analyses linking our findings to size regulation. For specifics, see response (7.1)

Minor Concerns:

Various aspects of the paper appear hastily constructed and lack sufficient attention to detail. We include a non-exhaustive list of examples below.

21. Reviewer Comment:

The omission of Geminin data beginning in Figure 3 due to “technical error” is concerning, especially because the authors go on to discuss the balance between division and cell growth in detail. It would seem a cell cycle marker would be particularly useful for the authors’ later analyses.

21.1. Author Response:

The unfortunate lack of data is the result of a human error in which the specific clone was forgotten in the incubator and was not added to the mix of clones on the day of the first (out of two) screen that we did. The magnitude of the experiment and the amount of data generated and analysed prevented us from repeating this screen just for this missing clone. Still, we have now added panel I in Sup figure 4, demonstrating the partition between the two clusters for all 12 reporters (including geminin) for all of the active drugs at the second screen. This is also now better explained in the legend of Figure 3.

22. Reviewer Comment:

2. Justification or experimental detail is lacking from various figures. For example, no justification is given for the selection of a bioactivity cutoff of 0.2 in the authors’ activity score; no sample sizes are offered in any of the data in Figure 2; the data in Figure 3 use only the 122 bioactive drugs, whereas the data in Figure 4 use the full set of 422 drugs; some images in Figure 1D appear to be scrambled.

22.1. Author Response:

We agree with the reviewer and apologies for the lack of clarity. We have now incorporated the following changes:

- Selection of a bioactivity cutoff of 0.2: In the revised manuscript, we have now included the figure below showing the distribution of the reporter activity scores in the DMSO control wells (n= 841) (Sup Fig. 2H). The 0.5 and 99.5 percentiles are -0.207 and 0.171 respectively. We have thus chosen this strict cutoff of +/-0.2 that represents less than 0.5% of false positive results.

- Sample sizes in Figure 2: we have now added the average cell counts per reporter. These numbers span between 530 and 656 cells/reporter.
- 122 bioactive drugs used in Figure 3 vs 422 drugs used in figure 4: The reason that the number of drugs was different between the figures is that in figure 3 we did not want the heatmap to be overloaded with data and thus only presented the 122 most active drugs. In figure 4 we used all drugs as it does not overload the presentation. To streamline the analysis we have now changes figure 4 to also only represent the same 122 active drugs. As the reviewer can see, this has only strengthened our main conclusion.
- Figure 1D: We are sorry for the mixup that was made by mistake by our graphic designer. We have now fixed the mixup of the panels.

23. Reviewer Comment:

3. Many of the plots throughout the paper are difficult to interpret because of the presence of hundreds of overlapping points. A data presentation with a heatmap of point density would be substantially more informative.

23.1. Author Response:

We thank the reviewer for his proposal. In figure 2C we have now changed the presentation to a density plot as suggested by the reviewer.

In figure 4a, we reduced the number of points from 422 to 122 (as the reviewer suggested above) which have also considerably reduced the overlapping points in the graph. We have also changed the opacity of the points which better represents the density.

24. Reviewer Comment:

4. Figure 6B abruptly changes cell lines without clear justification.

24.1. Author Response:

We apologize for not describing better in the text the reason for the switch. The reason for using multiple cell lines was to demonstrate universality of our observations, as opposed to a cell line-specific finding. In the revised manuscript we have extended our experiments to now include even more cell lines (e.g. A375, RPE1, HeLa, U2OS, SAOS2, 16HBE) and were happy to see that our main findings can be observed across all of these multiple cell lines, including some that are outside the context of malignancy. This is now also better clarified in the text.

Additionally, our followup experiments using RPE1 cells also provide the additional benefit of establishing a link with our previous studies on cell size [PMID: 29595474]. The morphology of RPE1 cells makes them particularly convenient for precise measurements on cell size. Since our measurements suggested mechanisms of cell size regulation, we shifted the study to a cell line where mechanisms of size regulation have been already carefully investigated.

Reviewers' comments:

Reviewer #1 (Remarks to the Author):

The authors have addressed all my comments satisfactorily.

Reviewer #2 (Remarks to the Author):

The authors have answered most of my initial questions and provided new data that strengthen the manuscript's conclusions. However, a few minor questions and comments remain.

1. One of my main concerns about the manuscript is the confusion created by the lack of distinction between true multiplexing at single cell resolution (e.g. all reporter readouts are from the same cell) and the here presented technique of cell population level multiplexing of live cell reporters (e.g. all reporters are from the same well but different cells) that enable dynamic assessment of signalling states. The title and abstract should clearly state that the here presented technique is about dynamic assessment of cellular signalling states using population level clonal multiplexing. This would avoid confusion and instead highlight the main strength and innovation of the presented technique.

2. A discussion of the advantages/disadvantages of the presented technique and true multiplexing at single cell resolution is essential. As just one example, clonal differences for each of the multiplexed reporters could lead to uncontrolled/undetected differences in response. Proper scoping of prior work, advantages and disadvantages of this work will increase its impact and help the community to interpret results.

3. The authors have provided the requested validation of the reporter readouts using western blotting (Supp Figure 2G). However, the current presentation of the data is inappropriate. Vertical cropping between bands that are directly compared should be avoided. Similarly loading controls should not be cropped vertically and included below the corresponding blots for the protein of interest.

4. Supp Fig 4I is not referenced in the text.

5. It is difficult to follow the different compound sets used in the different experiments. Figure 3 uses 122 active drugs according to their activity scores in A375. Supplementary Figure 4H uses 49 active drugs in PC9 and Supplementary Figure 4I uses 35 active drugs in A375. Details of the drug activity scores in the different experiments and cell lines should be included in the supplementary tables. In addition, information of the clustering results to the different compounds should also be included.

6. Supplementary Figure 5E is missing the annotation within the figure. In addition, Supplementary Figure 5F should be moved to supplementary Figure 6 to better reflect the flow of the manuscript.

7. The underlying data for Fig. 6E should be added as supplementary information to improve clarity of the data.

8. Detailed materials and methods for the analysis performed in Supplementary Figure 6 are missing. It's unclear how the drug influence scores are derived? Is each shown entry corresponding to a different drug?

9. A reference appears to be missing in the interpretation of Cluster C (page 13, line 7-9).

Addressing reviewer's comments

We would like to thank reviewer #2 for his/her additional comments which we have now fully addressed as can be seen below.

Reviewer #2 (Remarks to the Author):

The authors have answered most of my initial questions and provided new data that strengthen the manuscript's conclusions. However, a few minor questions and comments remain.

1. Reviewer Comment:

One of my main concerns about the manuscript is the confusion created by the lack of distinction between true multiplexing at single cell resolution (e.g. all reporter readouts are from the same cell) and the here presented technique of cell population level multiplexing of live cell reporters (e.g. all reporters are from the same well but different cells) that enable dynamic assessment of signalling states. The title and abstract should clearly state that the here presented technique is about dynamic assessment of cellular signalling states using population level clonal multiplexing. This would avoid confusion and instead highlight the main strength and innovation of the presented technique.

1.1 Author Response:

We have edited both the title and the abstract to make sure there is no confusion regarding the nature of the multiplexing.

The new title is now:

Visual barcodes for clonal-multiplexing of live microscopy-based assays

To the abstract we added the following description:

We then used visual barcodes to generate 'Signalome' cell-lines by mixing 12 clones of different live reporters into a single population, allowing....

We believe that these changes will prevent any misunderstanding regarding the nature of the Signalome cell lines.

2. Reviewer Comment:

A discussion of the advantages/disadvantages of the presented technique and true multiplexing at single cell resolution is essential. As just one example, clonal differences for each of the multiplexed reporters could lead to uncontrolled/undetected differences in response. Proper scoping of prior work, advantages and disadvantages of this work will increase its impact and help the community to interpret results.

2.1 Author Response:

As requested by the reviewer we have now added a paragraph to the discussion, clearly pointing out the advantages and disadvantages of the Signalome technology as compared to other multiplexing methods.

“While the signalome technology allows multiplexing live reporters in the same cell (Sup Figs. 1D,E), adding more than a very small number of such reporters per cell may have a substantial effect on the cells. Indeed, to the best of our knowledge, previous works that combined multiple live reporters in a single cell, have not exceeded a maximum of 5 reporters per cell.^{35,37-40} To achieve a much greater multiplexing, we mixed clones of cells, each containing a single reporter. While we demonstrate the use of 12-plex signalome cell lines we believe that much higher multiplexing can be easily achieved by this technology. In addition to the advantage in throughput by hi-plex reporter cell lines, as the readout of all reporters comes from the same wells, a substantial reduction in well-to-well variability is expected when comparing between the effect of perturbations on the different reporters. On the other hand, there are clear limitations for not comparing between reporters that are present in the same cells. The most obvious limitation is the difficulty to differentiate between events that occur in the same cell or in different cells. For example, if following a perturbation a subset of the cells of two clones that report on two signaling pathways show inhibition of the activity in these pathways, follow up experiments will need to be carried out to find out if inhibition of both pathways always accrue together in the same cells or maybe in different cells. In addition, as our signalome clones were frequently expanded from single cells, it is important to validate that the main findings also apply to the whole population and do not reflect a cell-specific phenotype. One way to overcome this limitation is that each signalome reporter cell line will originate from multiple cells rather than from a single one. It remains to be seen what effect of such an approach will have on the recall and precision rates

of the visual barcodes.”

3. Reviewer Comment:

3. The authors have provided the requested validation of the reporter readouts using western blotting (Supp Figure 2G). However, the current presentation of the data is inappropriate. Vertical cropping between bands that are directly compared should be avoided. Similarly loading controls should not be cropped vertically and included below the corresponding blots for the protein of interest.

3.1 Author Response:

We thank the reviewer for this comment, in the revised manuscript we added an annotation next to each band in Supp Figure 2G which reference the readers to bands in the original blots that are now added to the revised manuscript.

4. Reviewer Comment:

4. Supp Fig 4I is not referenced in the text.

4.1 Author Response:

We thank the reviewer for this remark, in the revised manuscript we referenced Supp Fig 4I in the text as well as in Figure 3 caption.

5. Reviewer Comment:

5. It is difficult to follow the different compound sets used in the different experiments. Figure 3 uses 122 active drugs according to their activity scores in A375. Supplementary Figure 4H uses 49 active drugs in PC9 and Supplementary Figure 4I uses 35 active drugs in A375. Details of the drug activity scores in the different experiments and cell lines should be included in the supplementary tables. In addition, information of the clustering results to the different compounds should also be included.

5.1 Author Response:

We thank the reviewer for this comment. Throughout the paper we used only two compound libraries which are clearly detailed in sup tables 1 and 2. The number of active drugs in each experiment was determined in an unbiased way from the activity of the reporters as clearly explained in both the main text and the methods section. In the revised manuscript we have now also added sup tables 5 and 6 that clearly show the reported activity score for each drug and the selection of active drugs from each library.

6. Reviewer Comment:

6. Supplementary Figure 5E is missing the annotation within the figure. In addition, Supplementary Figure 5F should be moved to supplementary Figure 6 to better reflect the flow of the manuscript.

6.1 Author Response:

We thank the reviewer for this comment, in the revised manuscript we added the annotation for Figure 5E and moved Figure 5F to supplementary Figure 6.

7. Reviewer Comment:

7. The underlying data for Fig. 6E should be added as supplementary information to improve clarity of the data.

7.1 Author Response:

As requested, data for Fig. 6E has now been added as Sup. table 7

8. Reviewer Comment:

8. Detailed materials and methods for the analysis performed in Supplementary Figure 6 are missing. It's unclear How the drug influence scores are derived? Is each shown entry corresponding to a different drug?

8.1 Author Response:

We thank the reviewer for this point. The plot title, "drug influence" was a mistake which we now replaced with the correct title, "Enrichment strength". In addition, we now added a section to the material and methods describing how "enrichment strength" score was derived and how the calculation in Sup Fig 6 was done. We have also revised the legend to the figure to better explain what was done.

9. Reviewer Comment:

9. A reference appears to be missing in the interpretation of Cluster C (page 13, line 7-9).

9.1 Author Response:

We thank the reviewer for this comment, in the revised manuscript we added the missing reference